# Polarization in Flood Risk Management?

## Sensitivity of norm perception and responsibility attribution to frequent flood experience

Lisa Köhler[1], Torsten Masson[2], Sungju Han[3], Christian Kuhlicke[3, 1]

[1]Institute of Environmental Science and Geography, University of Potsdam, Potsdam, Germany
[2]Institute of Psychology, Leipzig University, Germany
[3]Department of Urban and Environmental Sociology, Helmholtz-Centre for Environmental Research-UFZ Leipzig, Germany

*Correspondence to*: Lisa Köhler (lisa.koehler@uni-potsdam.de)

**Abstract.** In this study, we examine the relationship between frequent flood experience (FFE), norm perception, and responsibility attribution. Given that floods are assumed to occur more often in the future and that perceived norms and responsibility attribution are drivers of individual-level protective behavior against them, understanding these relationships is vital. The data for the current study come from a household survey conducted in flood-prone regions of the Federal State of Saxony (Germany) in 2020. We applied regression analyses to test for nonlinear relationships between FFE, responsibility attribution for flood risk management, and perception of social norms supporting private flood-protective behavior. In addition, we tested for moderating effects of these relationships. We identified four key findings. First, the relationship between frequent flood experience and responsibility attributions follows a nonlinear path. Changes in norm perceptions are less dynamic. Specifically, variations in effect strength and direction can only be observed for the perception of injunctive norms. Second, we detect a diverging trend among respondents who experienced multiple flood events, with greater responsibility attributed to public authorities and less to their own communities. Third, under consideration of interaction effects, we find increasing discrepancies in responsibility attributions and perception of social versus personal norms after the third flood event, depending on self-efficacy, control beliefs, and ingroup identification. Fourth, we observe a contradicting trend between perceived norms for protective behavior and responsibility attribution to the self/the community. These findings suggest a potential polarization in flood risk management, shaped by the perceived ability to manage floods and the social environment.

## 1   Introduction

Flooding is one of the most economically devastating natural hazards in Germany. The flash flood event in 2021 caused damages estimated at nearly 10.9 billion euros and claimed the lives of 190 people (GDV, 2024; Thieken et al., 2023). Individual-level protective behavior plays a crucial role in reducing flood risks (e.g., BMUV, 2022; UNISDR, 2015). Here, factors such as social norms, self-efficacy beliefs, risk perception, responsibility attributions, and negative affect are the most influential drivers of protection motivation (for recent reviews, see Bamberg et al., 2017; van Valkengoed & Steg, 2019). Furthermore, with global climate change altering precipitation patterns, flooding is likely to occur more frequently, putting more people and places at risk (IPCC, 2023). According to attribution studies by Kreienkamp et al. (2021) and Tradowsky et al. (2023), the 2021 flood event in Germany became more likely due to climate change. Aside from changes in hazard, increasing exposure, resulting from altering land use patterns and socioeconomic development, further adds to the risk of flooding (e.g., Bouwer, 2013; Merz et al., 2021; Steinhausen et al., 2022). Combining both the importance of individual-level protective behavior in reducing flood risks and expected changes in frequency and exposure to floods, it is obvious that we need to understand the sensitivity of protective behavior, and antecedents, to changes in the occurrence of floods.

Köhler and Han (2024) find that risk perception and coping appraisal are significantly related to the number of experienced flood events. Accordingly, more experienced people tend to associate a higher risk with flooding and greater effort with the implementation of protective measures. With the current study, we broaden this research scope to the social environment, specifically the perception of social norms and the attribution of responsibility in flood risk management (FRM). There is growing evidence that people are more likely to implement protective measures when they perceive this to be the social norm in their ingroup and when they feel responsible for reducing flood risk (e.g., Becker et al., 2015; Botzen et al., 2024; Botzen

et al., 2009b; Grothmann & Reusswig, 2006; Lo, 2013; Schrieks et al., 2024; Slotter et al., 2020; Soane et al., 2010). However, there is limited knowledge of the antecedents of responsibility attribution and norm perception. The current study addresses this gap by focusing on frequent flood experience (FFE) as a possible determinant.

In particular, we link FFE and responsibility attribution to different actors in FRM, namely, the self, the community, the city, and the state. With this detailed view, we hope to obtain insight into the dynamics of responsibility attribution over flood experience. Furthermore, we pinpoint a possible link between FFE and the perception of norms, with the aim of shedding light on the question of whether the experience of flooding may lead to collective learning processes and changes in perceived social norms. To get a better understanding of the observed links between frequent flood experience, norm perception, and responsibility attribution, we performed moderation analyses, taking expected helplessness during future floods, perceived self-efficacy to protect from future flooding, and identification with other residents as moderators. In Section 2, we discuss the existing literature on the relationship between hazard experience, responsibility attribution, and norm perception. This is followed by a description of the case study and the methods applied (Section 3). In the subsequent section, we present the results (Section 4), discuss them in Section 5, and draw conclusions from the current study in Section 6.

## 2    Literature review and research questions

### 2.1    Flood experience

Many studies that have integrated hazard experience into their analyses linked it to individual-level protective behavior (e.g., Duijndam et al., 2023; Kuhlicke et al., 2020a; Norris et al., 1999) or antecedents of behavior, such as risk perception (e.g., Botzen et al., 2015; de Wolf et al., 2023; Franceschinis et al., 2021; Lazrus et al., 2016; Mondino et al., 2020; Siegrist & Gutscher, 2008; see Bamberg et al., 2017 for meta-analytic evidence on the relationship between flood experience and risk perception or protective behavior). Whereas many studies suggest a positive relationship between experience and behavior, the significance and direction of the link also depend on the definition of experience. Common definitions are whether people have experienced at least one flood event or none (e.g., Heidenreich et al., 2020; Lawrence et al., 2014), having faced physical damages to private buildings (e.g., Osberghaus, 2017), having been evacuated (Botzen et al., 2009a; Reynaud & Aubert, 2020), or indirectly affected through family members and friends (Thistlethwaite et al., 2018). In the current study, we define flood experience as the frequency with which people have subjectively undergone such events (i.e. how often they report to have been affected by a flood event). While many studies have looked at the relationship between experience, individual-level protective behavior, and some associated motivational factors, studies investigating how hazard experience affects the perception of the social environment are scarce (also concluded in Begg et al., 2017).

### 2.2    Existing knowledge on norm perception and responsibility attribution in the context of flood risk

*Attribution of responsibility*

Previous studies suggest that there are several factors influencing whether people perceive themselves as responsible for flood protection. Rauter et al. (2020) find that persisting narratives from the past leading to strong trust in public flood protection and a lack of support for private adaptation hinder the acceptance of private responsibility. This aligns with Begg et al. (2017), concluding from their review of existing literature on justice and responsibility attribution in England, Germany, and the Netherlands that financial support by the state is necessary for local stakeholders to adopt responsibility in flood risk management. On the contrary, top-down planning processes, found to be (partially) prevalent in these countries, limit the local stakeholder's opportunities to get involved. Dillenardt et al. (2022) also observe that people feel more personally responsible if they perceive the financial support provided by the government for implementing protective measures as sufficient. Interviewing residents of flood-prone regions in England, Snel et al. (2021) find that they preferred public authorities to take more responsibility, exemplarily through monetary support for the implementation of flood protection measures. Snel et al. (2021) further conclude that respondents were not fully aware of the legal distribution of responsibility between stakeholders

involved in FRM (see also Johnson & Priest, 2008; Ommer et al., 2024). Henderson et al. (2025) infer from two Scottish case studies that an unclear distribution of responsibility can lead to its externalization from public institutions to communities and vice versa, a process referred to as the "cycle of externalization" (p. 13). They hypothesize that such externalization harms community resilience and slows down mitigation actions. From their survey among German residents affected by the 2021 flood, Ommer et al. (2024) observe that people either assign responsibility to themselves or to public authorities and rarely consider it a joint responsibility. Another potential factor influencing responsibility attribution is self-efficacy beliefs. Terpstra & Gutteling (2008) find higher self-efficacy beliefs to be a driver of ascribing responsibility to the self rather than to the government. Consequently, ongoing narratives, lack of (perceived) individual resources, governmental support, and unclear distribution of responsibility can challenge the acceptance of personal responsibility in FRM.

Studies linking flood experience and responsibility attribution yielded mixed results. Bichard & Kazmierczak (2012) observe a weak but significant positive correlation between being flooded before and ascribing responsibility for flood protection to the government. On the contrary, the more flood-experienced respondents in Kuang & Liao (2022) were more likely to ascribe responsibility to themselves. The authors suggest that this might be due to learning from past floods and the perception that only they can provide sufficient protection. Begg et al. (2017) find that people were more likely to agree that citizens are responsible if their experience was not severe. This aligns with the observed pattern by Terpstra & Gutteling (2008) that people with higher levels of expected control during a flood event are more likely to ascribe responsibility to the self, whereas those expecting higher consequences are more likely to ascribe it to the state. These two studies suggest that the attribution of responsibility to flood-prone residents is related to their perceived ability to deal with floods. Lastly, Bubeck et al. (2020) find that the ascription of responsibility to the self versus public authorities remains widely stable over the years after a flood event.

To conclude, there are some studies that have looked at the interplay between hazard experience and responsibility attribution. However, the role played by hazard *frequency* is widely missing. Most of these studies have examined responsibility attribution to either the self or the government. With the current study, we broaden the research lens by also integrating responsibility attributed to the community (i.e., local ingroup) into our analyses.

*Perception of social norms*

Social norms can be classified into descriptive and injunctive norms. Descriptive norms are defined as the (perceived) actual behavior of others (APA, 2025a), and injunctive norms as the perception of how people should behave (APA, 2025e).

Psychological research has demonstrated that individuals use different information sources to infer social norms (e.g., on private flood-protective behavior), including information about the respective behaviors and attitudes of significant others (Tankard & Paluck, 2016). Therefore, the perception of social norms may be subject to biases in information search and processing (e.g., Jonas et al., 2006; for models on risk information-seeking behavior see Griffin et al., 1999; Kahlor, 2010). For example, individuals who identify themselves highly with their social ingroup (e.g., their local community) might be motivated to perceive ingroup norms that support a positive image of their group (see Masson et al., 2016, for experimental evidence on group-serving norm perceptions among high identifiers). Although social norms are a strong predictor of protective behavior (for a review, see van Valkengoed & Steg, 2019), little research has examined the factors that shape them. In the following, we briefly discuss studies that have explored the link between risk perception, coping appraisal, and social norms to gain insights into their role in FRM.

Bubeck et al. (2018) observe that people who assume that relevant others undertake actions to protect from flooding are more likely to perceive themselves as *self-effective* (see also Kurata et al., 2022; McIvor & Paton, 2007; Seebauer & Babcicky, 2020). Seebauer & Babcicky (2020) hypothesize that social norms might particularly influence self-efficacy when the corresponding behavior is visible to others. This aligns with Bandura (1997, p. 87) that observing (similar) others successfully

performing a specific behavior can strengthen the observer's self-efficacy in being capable of carrying out the respective behavior as well. Further, Bubeck et al. (2018) find that the perception of others undertaking protective measures negatively relates to the *expected effort* to implement non-structural measures and positively to the perceived *effectiveness* of purchasing insurance and implementing structural- and non-structural measures (see also McIvor & Paton, 2007; Vinnell et al., 2019; Wood et al., 2012). De Wolf et al. (2023) observe that perceived injunctive norms for flood protection positively relate to individual *risk perception*, supported for perceived descriptive norms by Kurata et al. (2022). Lo (2013) finds that injunctive and descriptive norms mediate the relationship between risk perception and the purchase of flood insurance, concluding that people are more likely to act on their risk perception when they perceive protective behavior to be the social norm. They consider the relationship between perceived social norms and risk perception to be bidirectional: people might perceive others to be acting because they appraise flooding as a serious threat, or people might be more likely to perceive flooding as a risk because they observe others undertaking protective measures (see also Social Amplification of Risk; e.g., Kasperson et al., 1988).

The discussion of the existing literature on the development of social norms in the context of private flood protection shows that there is already some knowledge. Exemplarily, social norm perception and perceived self-efficacy seem to be interrelated. The same applies to the link between social norm perception and threat appraisal. However, the effect of personal experience, exemplary through social learning and interaction with other members of the ingroup, remains relatively unclear.

## 2.3    Linking flood experience, norm perception, and attribution of responsibility theoretically

Although there are studies examining the drivers of norm perception and responsibility attribution in the domain of flood risk management, studies investigating personal experience as driver, especially of norm perception, are scarce. In addition, the relationships between experience, perceived norms, and responsibility attribution are rarely integrated in existing theoretical frameworks. In the following section, we attempt to theoretically reason the hypothesized effect of frequent flood experience on the development of perceived norms and responsibility attribution by combining existing theories and concepts.

Developments in norm perception and responsibility attribution over frequent flood experience could result from observing or adopting actual changes in significant others' behaviors and attitudes (see Theory of Social Learning, Bandura, 1973, p.68). Fischer et al. (2013, p.127) state that it is particularly social norms that are often acquired through social interaction. Frequent flood experience and associated exchanges, such as during joint reconstruction activities, might provide these opportunities for observation, interaction, and learning from each other (Kuang & Liao, 2020; van der Linden et al., 2015). Moreover, it has been shown that especially in uncertain situations, informational social influence is vital in shaping opinions (Fischer et al., 2013, p.143; Sherif, 1935). In addition to learning from others, personal experience might contribute to a more realistic evaluation of the effectiveness and limitations of private precautions, thus impacting responsibility attribution.

Another factor causing shifts in norm perception and responsibility attribution following frequent flood experience may be rooted in motivated cognition. The concept of motivated cognition refers to the adjustment of perceptions to avoid unfavorable states of mind (e.g., Fischer & Wiswede, 2009, p. 214-215). One of such unfavorable states could be diminished self-esteem or increasing uncertainty when experiencing multiple floods. According to the Social Identity Theory (SIT), an individual's perceptions of their ingroup shape their social identity (Tajfel & Turner, 1979, 1986), which is connected to their self-concept (Tajfel & Turner, 1986). This is especially the case if people highly identify with the respective group (e.g., Fritsche et al., 2013; Masson et al., 2016; Rothbaum et al., 1982; Stollberg et al., 2015). In the context of the current study could the analogy suggest that attributing positive traits to the ingroup, such as a consensus on engaging in flood-preventive actions (i.e., social norms), can be a leverage to maintain a competent self-concept and regain a sense of control when undergoing multiple flood events, which is found to be negatively linked to feelings of control and coping appraisal (Köhler and Han, 2024). Attributing responsibility to public authorities rather than oneself could further serve as a way to rationalize being flooded frequently

despite adaptation, without questioning one's own ability but rather blaming limits of private adaptation (e.g., Cologna et al., 2017; Gosling et al., 2006).

In fact, both factors, actual changes in behaviors and attitudes of others and motivated cognition, can shape people's image of their social environment (Frey, 1997; p.58).

## 3    Research questions of the current study

The literature review (2.2) revealed that there is only little knowledge of the relationship between frequent flood experience, responsibility attribution, and especially norm perception. From this gap, we derive the following overarching research question (RQ1) of the current study:

**RQ1:** How does the attribution of responsibility for flood protection to different actors (individuals, the community, the city, and the state) and the perception of social norms relate to frequent flood experience?

To understand the patterns observed in RQ1 and link them theoretically, we exploratively tested for boundary conditions for the effect of FFE on perceived norms and responsibility attributions. Here, we hypothesized that perceived control of future flood risk and group identification could play a mediating role. This hypothesis is reasoned in the theoretical considerations presented in section 2.3. We formulated the following research question:

**RQ2:** Do factors related to perceived control of future flood risk (i.e., helplessness and self-efficacy) and social connectedness
(i.e., group identification) moderate the relationship between frequent flood experience, norm perception, and responsibility attribution?

With the third research question, we aim to uncover possible discrepancies between the respondents' perceived social norms for flood protection and respective responsibility attributions:

**RQ3:** How do changes in perceived norms and attributions of responsibility interact in the context of frequent flood
experience?

## 4    Case study and methodology

### 4.1    Study site, data collection and sample characteristics

The data for the analyses stem from a household survey, carried out as part of the PIVO Project in 2020/2021 in 11 randomly chosen communities in the Federal State of Saxony (Germany, Siedschlag et al., 2023; see Fig.1). For the current study, we
used the 2020 survey wave. In total, 1.884 people participated in the survey (response rate: 59.3%). Most of the respondents had not experienced any flood event before, while almost 17% indicated having experienced at least three floods (Fig. 2). The Federal State of Saxony has been impacted by several flood events in the past, among which the 2002 flood event was the most severe: 21 people died and flood damage reached 8.6 billion euros (DKKV, 2015; LTV, 2022). Other devastating floods occurred in 2010 and 2013 (LTV, 2022).

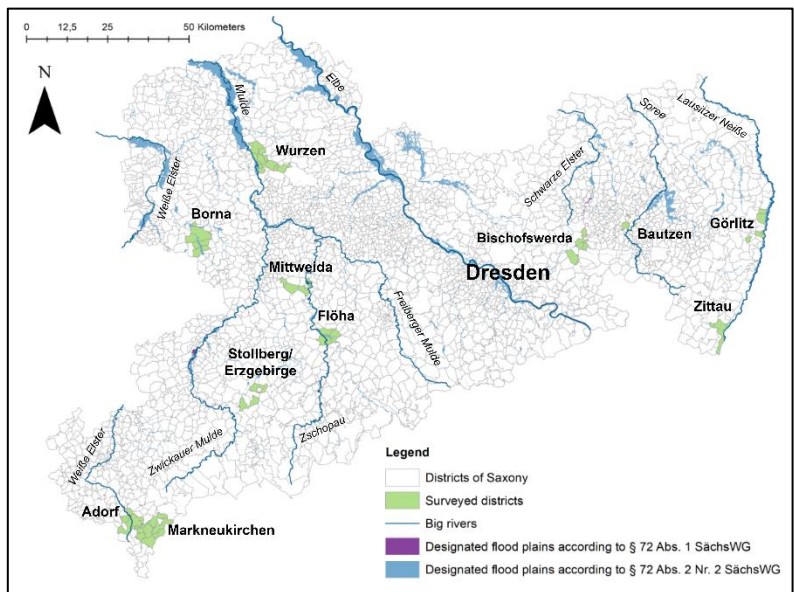

**Figure 1:** **Federal state of Saxony with surveyed districts highlighted, designated flood plains, and main rivers; created with ArcGIS, layer sources: districts of Saxony: GeoSN - Landesamt für Geobasisinformation Sachsen, 2024; big rivers: Geofabrik, 2024; designated flood plains: LUIS - Landwirtschaft- und Umweltinformationssystem für Geodaten, 2024**

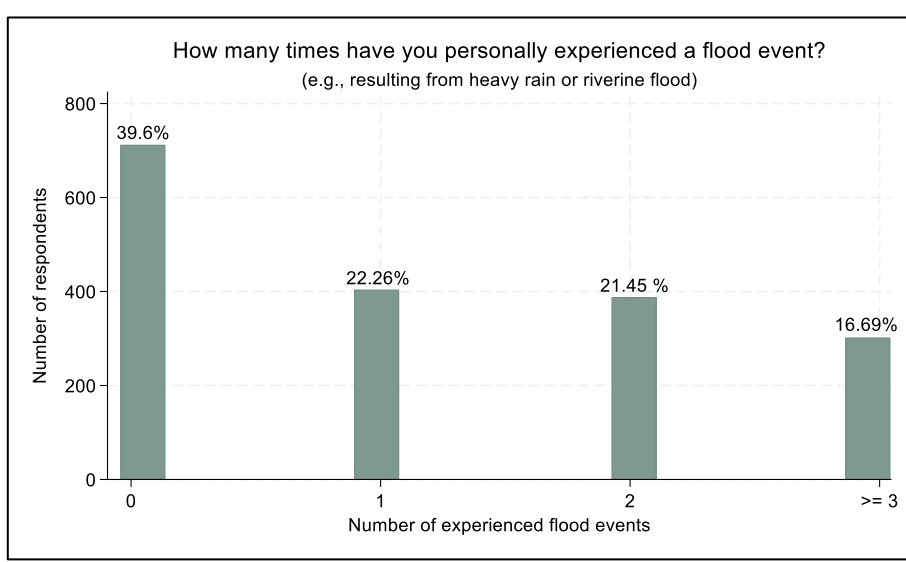

**Figure 2:** **Distribution of flood experience in the dataset**

The survey participants are on average older, more likely to own their place of residence, and more educated compared to the general population of the Federal State of Saxony (Table 1). Particularly the differences in the ownership structure might harm the representativeness of the sample regarding the attributions of responsibility. However, this assumption must be seen against the backdrop of limited existing knowledge on determinants of responsibility attribution for flood risk management, exemplary 205 the effect of ownership.

**Table 1: Characteristics of the sample and the federal state**

|  | **Pivo Sample 2020** | **Federal State of Saxony** |
|---|---|---|
| **Mean age** | 57.7 years | 46.9 years (2020; Statista, 2025) |
| **Gender** (M/ F/ D) | 53.5 % / 46 % / 0.5 % | 49 % / 51 % / not indicated (STLA, 2025a, p.6) |
| **Ownership** (renters/ owners) | 25.4 % / 74.6 % | 59.9% / 30.9% (STLA, 2025b, p.8) |

| Education level (A-level/ Hauptschule, Volksschule/ Mittlere Reife, Realschule, POS/ no degree/ other | 36.8% /11.9 % / 49.01 % / 0.62 % / 1.58% | 29.4 %/ 15.1 %/ 48.1 % / 4 % / no information (STLA, 2025a, p.7) |
|---|---|---|
| Building characteristics | Detached single-family house: 48.4% Semi-detached house: 6.99% Terraced house: 7.16% Multi-family house: 29.7% Other: 7.72% | Detached single- or multi-family house: 45.8 % Semi-detached house: 7.8 % Terraced house: 39.9 % Other: 6.5 % (STLA, 2025b, p.6) |

## 4.2 Selected variables

The selected variables for the analyses are presented in Table 2 (for a correlation matrix, see Appendix A). Except for the variable indicating the number of experienced flood events, all variables are scaled from 1 to 7, based on the 7-point-likert scale. The Likert-scale is a commonly applied approach to measure perceptions and attitudes in psychological research. The measurements of the variables were adapted from previous studies (for comparison, see Heidenreich et al. 2020, Leach et al., 2008, Masson et al., 2019). For a better understanding of the applied psychological constructs and their operationalization, see the definitions presented in Table 3.

**Table 2: Variables and summary statistics**

| Variable | Question(s)/Statement(s) in the questionnaire [Values] | Summary statistics, standard deviation in brackets | Role in the analysis |
|---|---|---|---|
| **Injunctive** norm perception (***inj***, composite variable of two questions) | "Most people in my city expect me to protect myself from flooding." [1= strongly disagree; 7= strongly agree] | Mean=4.23 (1.93), N= 1680 | Dependent variable |
| | "Most people in my city think that everyone should protect themselves." [1= strongly disagree; 7= strongly agree] | Mean=4.05 (1.85), N=1682 | |
| **Descriptive** norm perception (***des***, composite variable of two questions) | "Most people in my city undertake protective measures privately." [1= strongly disagree; 7= strongly agree] | Mean=4 (1.7), N= 1682 | Dependent variable |
| | "Most people in my city implement precautionary measures." [1= strongly disagree; 7= strongly agree] | Mean=3.92 (1.66), N=1683 | |
| **Personal** norm (***pers***, composite variable of two questions) | "I feel morally obliged to participate in collective actions." [1= strongly disagree; 7= strongly agree] | Mean=3.6 (1.95), N=1783 | Dependent variable |
| | "I have a bad conscience when not contributing to collective actions." [1= strongly disagree; 7= strongly agree] | Mean=3.24 (1.99), N= 1729 | |
| Perceived **responsibility** of the **individual** | "I am personally responsible for reducing flood damage." [1= strongly disagree; 7= strongly agree] | Mean=3.02 (1.98), N=1698 | Dependent variable |
| Perceived **collective responsibility** of the community | "We, the people of my city, are collectively responsible for reducing flood damage." [1= strongly disagree; 7= strongly agree] | Mean=3.75 (1.8), N=1701 | Dependent variable |
| Perceived **responsibility** of the **city**\* | "The city is responsible for reducing flood damage." [1= strongly disagree; 7= strongly agree] | Mean=5.51 (1.51), N=1746 | Dependent variable |
| Perceived **responsibility** of the **state**\* | "The state is responsible for reducing flood damage." [1= strongly disagree; 7= strongly agree] | Mean=5.8 (1.43), N=1732 | Dependent variable |

| | | | |
|---|---|---|---|
| **Frequent flood experience** (***FFE***) | "How many times have you personally experienced a flood (heavy rain or riverine flood)?" [0=none; 3=at least three] | Mean=1.15 (1.12), N=1851 | Explanatory variable |
| Perceived **self-efficacy** (composite variable of three questions) | "I trust myself to be able to protect my home from flooding by implementing precautionary measures." [1= strongly disagree; 7= strongly agree] | Mean=3.93 (2), N=1742 | Moderator variable, conceptualized as perceived control |
| | "I can take precautions against flood damage." [1= strongly disagree; 7= strongly agree] | Mean=4.04 (2.04), N=1758 | |
| | "I can prepare well for future floods." [1= strongly disagree; 7= strongly agree] | Mean=3.67 (1.96), N=1746 | |
| **Identification** with people living in the same city | "I identify with people in my city." [1= strongly disagree; 7= strongly agree] | Mean=4.79 (1.66), N=1810 | Moderator variable, conceptualized as social connectedness |
| Expected **helplessness** during the next flood event | "When I think about future flood events, I feel helpless." [1= strongly disagree; 7= strongly agree] | Mean=3.65 (1.96), N=1821 | Moderator variable, conceptualized as perceived control |

\* Throughout the paper, we use the terms "public authorities" or "public actors" to refer to the city/state

By integrating the variables "self-efficacy" and "helplessness" as moderators in the analysis, we aim to shed light on the role of perceived control in shaping norm perception and responsibility attribution when individuals experience multiple flood events. This subsumption under perceived control is reasoned in both factors emphasizing the (perceived) capacity to shape a certain situation or associated outcomes through own agency (see Table 3). Helplessness specifically refers to a state during a threatening situation. Self-efficacy, in turn, highlights the perceived ability to undertake measures to avoid such a situation. To obtain interpretable classes for the moderation analyses, we built three groups for each moderator and assigned respondents based on their responses to the statements (Table 4).

**Table 3: Definitions of applied constructs and variables**

| | Definition |
|---|---|
| Experience | "an event that is actually lived through, as opposed to one that is imagined or thought about." (APA, 2025b) |
| Social norm | "any of the socially determined consensual standards (…)" (APA, 2025g) |
| Injunctive norm perception | "any of various socially determined consensual standards (social norms) that describe how people should act, feel, and think in a given situation, irrespective of how people typically respond in the setting." (APA, 2025e) |
| Descriptive norm perception | "any of various consensual standards (social norms) that describe how people typically act, feel, and think in a given situation." (APA, 2025a) |
| Personal norm | "sense of obligation to take proenvironmental actions" (Stern, 2000; p. 412) |
| Group identification | "the act or process of associating oneself with a group and its members, such that one imitates and internalizes the group's actions, beliefs, standards, objectives, and so forth." (APA, 2025c) |
| Helplessness | "A state of incapacity, vulnerability, or powerlessness associated with the perception that one cannot do much to improve a negative situation that has arisen." (APA, 2025d) |
| Self-efficacy | "An individual's subjective perception of their capability to perform in a given setting or to attain desired results." (APA, 2025f) |
| Perceived control | "Subjective belief that one can achieve desired sates and avoid undesired states through one's own efforts." (Dorsch, 2025; translated from German) |

**Table 4: Number of respondents per group (moderator analyses)**

| Moderator | Self-efficacy | Group identification | Helplessness |
|---|---|---|---|
| **Low** [answer: 1,2] | 440 (25.6%) | 191 (10.6%) | 638 (35%) |

| | | | |
|---|---|---|---|
| **Medium** [answer: 3,4,5] | 919 (53.5%) | 911 (50.3%) | 775 (42.6%) |
| **High** [answer: 6,7] | 360 (20.9%) | 708 (39.1%) | 408 (22.5%) |

### 4.3 Analytical steps

Since the norm variables reflect the average of two variables measuring the underlying construct (see Table 2), we z-standardized the norm and responsibility variables to make the effects of FFE comparable. In the following, we present the analytical steps for each research question separately.

**RQ1: How does the attribution of responsibility for flood protection to different actors (individuals, the community, the city, and the state) and the perception of social norms relate to frequent flood experience?**

We applied single-level linear regression models with FFE as the only independent variable. For responsibility attribution, the descriptive analyses suggest a nonlinear relationship, which we accounted for by estimating the effect of each flood experience individually. Exemplarily, formula 1 presents the equation for investigating the effect of the first flood experience taking zero flood experience as the reference category. For the norm constructs, the descriptive analyses suggest rather linear relationships, reasoning the linear model specification presented in formula 2.

$$z\_RESPONSIBILITY_j(reference: FFE\_0) = \beta_0 + \beta_1 * FFE\_1 + \beta_2 * FFE\_2 + \beta_3 * FFE\_3 + \varepsilon_j \tag{1}$$

$$z\_NORM_j(FFE) = \beta_0 + \beta_1 * FFE + \varepsilon_j \tag{2}$$

$z\_RESPONSIBILITY_j = Standardized\ value\ of\ responsibility\ attribution\ for\ respondent\ j$

$z\_NORM_j = Standardized\ value\ of\ perceived\ norm\ for\ respondent\ j$

$$FFE\_0 = \begin{cases} 1, & if\ respondent\ j\ has\ not\ experienced\ any\ flood\ event \\ 0, & if\ respondent\ j\ has\ experienced\ at\ least\ one\ flood\ event \end{cases}$$

$$FFE\_1 = \begin{cases} 1, & if\ respondent\ j\ has\ experienced\ one\ flood\ event \\ 0, & if\ respondent\ j\ has\ experienced\ less\ or\ more\ than\ one\ flood\ event \end{cases}$$

$$FFE\_2 = \begin{cases} 1, & if\ respondent\ j\ has\ experienced\ two\ flood\ events \\ 0, & if\ respondent\ j\ has\ experienced\ less\ or\ more\ than\ two\ flood\ events \end{cases}$$

$$FFE\_3 = \begin{cases} 1, & if\ respondent\ j\ has\ experienced\ at\ least\ three\ flood\ events \\ 0, & if\ respondent\ j\ has\ experienced\ less\ than\ three\ flood\ events \end{cases}$$

We further analyzed whether the effect of FFE on norm perception/responsibility attribution differs between the distinct constructs (e.g., whether the perception of descriptive norm develops statistically significantly from the personal norm). Therefore, we long-transformed the dataset, obtaining multiple rows per respondent $j$, in which the norm/responsibility measurements are nested. As observations from the same person are not independent, we applied a mixed-effects model with random intercepts. To compare the effects of flood experience between different norm/responsibility constructs, we added $n_{ij}\ and\ r_{ij}$ as interaction effect (formulas 3,4).

$$z\_NORM_{ij}(reference: FFE\_0) = \beta_0 + \beta_1 * FFE\_1 + \beta_2 * FFE\_2 + \beta_3 * FFE\_3 + \beta_4 * n_{ij} +$$

$$\beta_5 * FFE\_1 * n_{ij} + \beta_6 * FFE\_2 * n_{ij} + \beta_7 * FFE\_3 * n_{ij} + u_j + \varepsilon_{ij} \tag{3}$$

$$z\_RESPONSIBILITY_{ij}(reference: FFE\_0) = \beta_0 + \beta_1 * FFE\_1 + \beta_2 * FFE\_2 + \beta_3 * FFE\_3 +$$
$$\beta_4 * r_{ij} + \beta_5 * FFE\_1 * r_{ij} + \beta_6 * FFE\_2 * r_{ij} +$$
$$\beta_7 * FFE\_3 * r_{ij} + u_j + \varepsilon_{ij} \tag{4}$$

$$z\_NORM_{ij} = \begin{cases} z\_injunctive\ norm, & if\ n_{ij} = 1 \\ z\_descriptive\ norm, & if\ n_{ij} = 2 \\ z\_personal\ norm, & if\ n_{ij} = 3 \end{cases}$$

$$z\_RESPONSIBILITY_{ij} = \begin{cases} z\_individual\ responsibility, & if\ r_{ij} = 1 \\ z\_collective\ responsibility, & if\ r_{ij} = 2 \\ z\_city\acute{}s\ responsibility, & if\ r_{ij} = 3 \\ z\_state\acute{}s\ responsibility, & if\ r_{ij} = 4 \end{cases}$$

$n_{ij}$= nested norm type i (e.g., injunctive norm) for respondent j

$r_{ij}$= nested responsibility type i (e.g., ascribed individual responsibility) for respondent j

$u_j$ = respondent-specific (=level 2) random intercept effect

$e_{ij}$= norm-/responsibility-type (=level 1) random effect

The decision to not integrate further explanatory variables, such as sociodemographic characteristics, is predominantly reasoned in the research objective of the current study. The primary aim is to investigate potential relationships between FFE, norm perception, and responsibility attribution instead of specifying a model that explains a huge share of the variance in the outcome variables. Further, there is (to our knowledge), no theoretical framework nor substantial empirical evidence

suggesting that sociodemographic variables play a crucial role in norm perception and responsibility attribution for flood risk management, reasoning their integration in the regression model[1].

**RQ2: Do factors related to perceived control of future flood risk (i.e., perceived helplessness, self-efficacy beliefs) and social connectedness (i.e., group identification) moderate the relationship between frequent flood experience, norm perception, and responsibility attribution?**

To test for moderating effects of the relationship between responsibility attribution, norm perception, and flood experience, we added interaction terms to the formula (1):

$$z\_Y_j(reference\ category: FFE\_0) = \beta_0 + \beta_1 * FFE\_1 + \beta_2 * FFE\_2 + \beta_3 * FFE\_3 + \beta_4 * m_j +$$
$$\beta_5 * FFE\_1 * m_j + \beta_6 * FFE\_2 * m_j + \beta_7 * FFE\_3 * m_j + \varepsilon_j \tag{5}$$

$m_j$ = value of the moderator m for the respondent j

We further added moderators to formula (3) and (4) to test whether the moderation effects differ between distinct norm and responsibility constructs, leading to formula (6) and (7):

$$\_z\_NORM_{ij}(reference: FFE\_0) = \beta_0 + \beta_1 * FFE\_1 + \beta_2 * FFE\_2 + \beta_3 * FFE\_3 + \beta_4 * m_j +$$
$$\beta_5 * n_{ij} + \beta_6 * FFE\_1 * m_j * n_{ij} + \beta_7 * FFE\_2 * m_j * n_{ij} +$$
$$\beta_8 * FFE\_3 * m_j * n_{ij} + u_j + \varepsilon_{ij} \tag{6}$$

---

[1] Integrating age, gender and ownership structure as covariates to the model does not alter the effect strength and significance of FFE. We observe that homeowners are statistically significantly less likely to attribute responsibility to the city and more likely to attribute responsibility to themselves.

$$z\_RESPONSIBILITY_{ij}(reference: FFE\_0) = \beta_0 + \beta_1 * FFE\_1 + \beta_2 * FFE\_2 + \beta_3 * FFE\_3 + \beta_4 * m_j +$$
$$\beta_5 * r_{ij} + \beta_6 * FFE\_1 * m_j * r_{ij} + \beta_7 * FFE\_2 * m_j * r_{ij} +$$
$$\beta_8 * FFE\_3 * m_j * r_{ij} + u_j + \varepsilon_{ij} \tag{7}$$

To account for the different sample sizes per category of the moderators and potential biases due to outliers (see Table 4), we applied additional analyses, incorporating heteroskedasticity-robust standard errors into the models (Huber-White estimator). For the simple OLS (ordinary least squares regression, formula (5)), we applied the HC3 correction. For the mixed-effects model (formulas (6) and (7)), we employed HC1, as HC2 and HC3 are not applicable. For some interactions, the coefficients become insignificant following these adjustments. However, since the p-values only slightly exceed the chosen significance threshold of $p<0.05$, we decided to still interpret these coefficients alongside those that remain statistically significant. For transparency, the regression coefficients and 95% confidence intervals for both estimations are presented in the Appendix B and C.

**RQ3: How do changes in perceived norms and attributions of responsibility interact in the context of frequent flood experience?**

To answer RQ3, we adjusted the mixed-effects model (formulas (3) and (4)). Instead of analyzing norm perceptions and responsibility attributions separately, we combined them into a single model by introducing the interaction variable $nr_{ij}$ and the outcome variable $z\_NORM\_RESPONSIBILITY_{ij}$ (formula 8). The variable $nr_{ij}$ specifies which norm or responsibility construct serves as the reference category against which the remaining constructs are compared. This model setup allows us to contrast how FFE interacts with norm perceptions versus responsibility attributions within the same model, enabling us to draw conclusions on statistically significant differences between them. We iteratively compared single norm constructs to all responsibility variables.

$$z\_NORM\_RESPONSIBILITY_{ij}(reference: FFE\_0) = \beta_0 + \beta_1 * FFE\_1 + \beta_2 * FFE\_2 + \beta_3 * FFE\_3 + \beta_4 * nr_{ij} +$$
$$\beta_5 * FFE\_1 * nr_{ij} + \beta_6 * FFE\_2 * nr_{ij} + \beta_7 * FFE_3 * nr_{ij} + u_j +$$
$$\varepsilon_{ij} \tag{8}$$

$$z\_NORM\_RESPONSIBILITY_{ij} = \begin{cases} z\_injunctive\ norm, \ if\ nr_{ij} = 1 \\ z\_descriptive\ norm, \ if\ nr_{ij} = 2 \\ z\_personal\ norm, \ if\ nr_{ij} = 3 \\ z\_individual\ responsibility, \ if\ nr_{ij} = 4 \\ z\_collective\ responsibility, \ if\ nr_{ij} = 5 \\ z\_city\acute{}s\ responsibility, \ if\ nr_{ij} = 6 \\ z\_state\acute{}s\ responsibility, \ if\ nr_{ij} = 7 \end{cases}$$

## 5    Results

### 5.1    Trends in responsibility attribution and perceived norms for flood protection over flood experience (RQ1)

*Responsibility attribution*

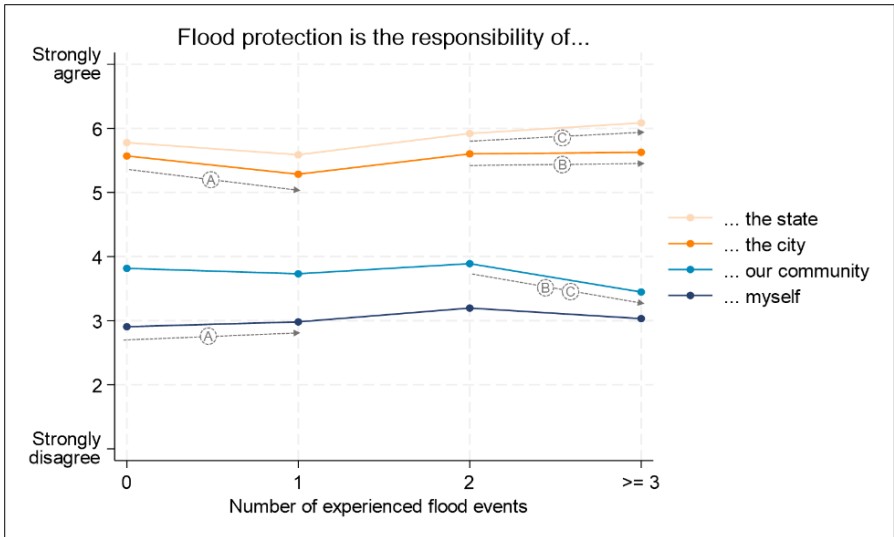

**Figure 3:**          Development of ascription of responsibility over flood experience; statistically significant differences in the effect of experience between different responsibility attributions are marked with the same letter; capitalized letter: p<0.01

Figure 3 hints at the existence of nonlinear relationships between FFE and responsibility attribution. Accordingly, the attributed responsibility to public authorities decreases after the first flood event, followed by an increase after the second flood (statistically significant, see Table 5). Further, after the third flood event, respondents tend to ascribe significantly less responsibility to the community (see Table 5). We conclude an increasing gap between responsibility attributed to public authorities vs. the community, supported by statistically significantly differences in the effects of the third flood event (Fig. 3, B, C). Personal responsibility remains widely stable (Fig. 3), supported by the insignificant regression coefficients presented in Table 5.

**Table 5: Linear/non-linear effects of flood experience on the ascription of responsibility**

| Effect of the … | Personal responsibility (standardized) | Collective responsibility (standardized) | City's responsibility (standardized) | State'sresponsibility (standardized) |
|---|---|---|---|---|
| first flood experience | 0.05 | -0.05 | -0.19** | -0.13* |
| second flood experience | 0.11 | 0.09 | 0.22** | 0.25** |
| third flood experience | -0.08 | -0.24** | 0.01 | 0.1 |
| Adjusted R-squared | 0.002 | 0.005 | 0.006 | 0.01 |
| N | 1647 | 1678 | 1721 | 1707 |
| F-statistic | 2 | 3.63* | 4.36** | 7.95** |

∗ p < 0.05; ∗∗ p < 0.01

*Norm perception*

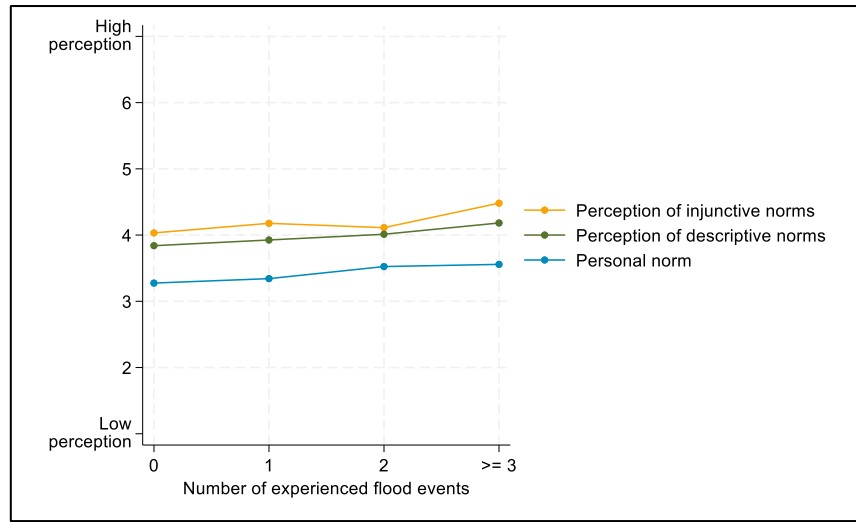

**Figure 4:**         **Development of norm perception over flood experience**

From Figure 4, we conclude that the more floods people have experienced, the more likely they are to believe that others expect them to undertake protective actions (injunctive norms), that others implement private adaptation measures (descriptive norms), and to feel morally obligated to participate in collective actions to protect from flooding (personal norms). This is supported by the significant regression coefficients, presented in Table 6. Only for the perception of injunctive norms can we find signals for a nonlinear relationship. Contrarily to the development of responsibility, are changes in norm perception over FFE more stable and do not statistically significant differ in their development over flood experience.

**Table 6: Linear and non-linear effects of flood experience on the perception of norms**

| Effect of the … | Perception of injunctive norms (standardized) | Perception of descriptive norms (standardized) | Personal norms (standardized) |
|---|---|---|---|
| first flood experience | 0.08 | 0.05 | 0.04 |
| second flood experience | -0.04 | 0.07 | 0.1 |
| third flood experience | 0.23** | 0.1 | 0.02 |
| Linear effect of frequent flood experience | 0.07** | 0.07** | 0.06** |
| Adjusted R-squared | 0.007 | 0.005 | 0.003 |
| N | 1638 | 1642 | 1692 |
| F-statistic | 4.86** | 3.45* | 2.95* |

* $p < 0.05$; ** $p < 0.01$

### 5.2     Moderation effects on the observed relationships (RQ2)

Building on the diverging developments in responsibility attribution and (slightly) norm perception after people had experienced three or more flood events (RQ1), we tested for moderating effects based on the theories discussed, aimed at better understanding these patterns. Therefore, in the subsequent description of the results, we primarily focus on the development after the third flood event and whether the diverging developments are pronounced in specific groups.

*Attribution of responsibility*

The moderation analysis with helplessness associated with future floods did not reveal statistically significant differences after the third flood event, thus not improving our understanding of the patterns observed in RQ1. Therefore, we excluded it from the subsequent description of the results.

## Self-efficacy beliefs

There is a diverging development between the ascription of responsibility to the self/the community vs. to public authorities for people with high self-efficacy beliefs (Fig. 5). The decrease in the personally felt responsibility after the third flood event differs significantly from the developments in the other groups (Fig. 5, left, a, B). The increase in ascribed responsibility to the city differs significantly from the development among respondents with low self-efficacy beliefs (Fig. 5, left, d). The augmenting gap between responsibility attributed to the self/community vs. public authorities after the third flood differs statistically significantly from the respective developments in the low and medium groups (Fig. 5, right, A-E).

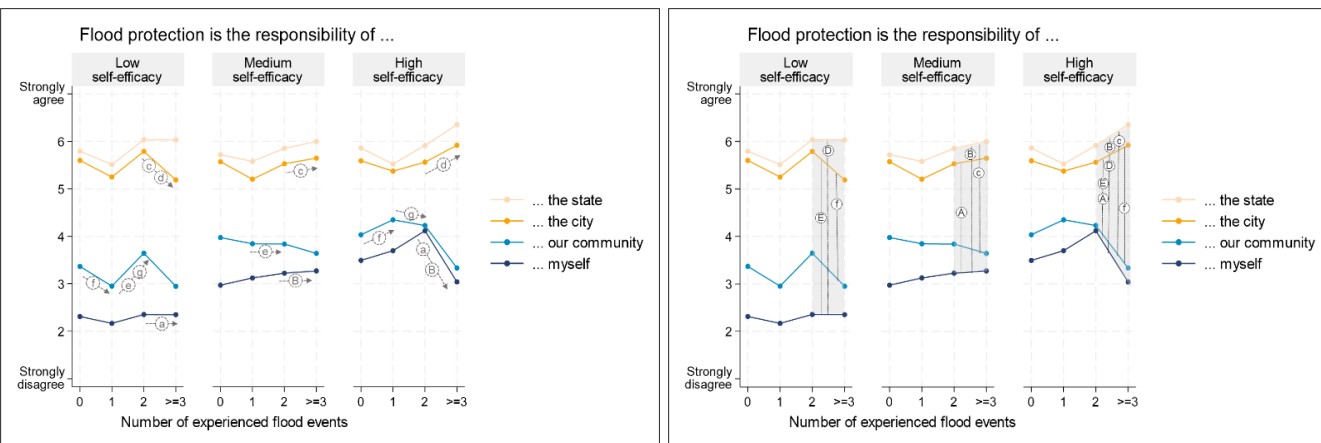

**Figure 5:** Development of ascription of responsibility over flood experience, depending on self-efficacy beliefs. Left: Statistically significant differences in the effect of flood experience are marked by the same letter, e.g., letter B: the decrease in the personally felt responsibility after the third flood event in the high effective group differs statistically significantly from the slight increase in the medium effective group. Right: Statistically significant differences in the diverging development of two responsibility variables are marked with the same letter, e.g., letter B: the increasing difference between the ascribed responsibility to the state and the individual after the third flood event in the high effective group differs statistically significantly from the respective development in the medium effective group. Lowercase letter: p<0.05, capitalized letter: p<0.01; l; applying heteroskedasticity-consistent standard errors: left: c becomes insignificant (HC3); right: f becomes insignificant (HC1)

## Strenght of ingroup identification

The analyses with ingroup identification as moderator indicate a divergent trend in responsibility attribution among respondents who reported strong ingroup identification (Fig. 6). The decrease in the personally felt responsibility after the third flood event differs statistically significantly from the respective increase among low-identifiers (Fig. 6, left, b). The same applies to the augmenting gap between responsibility attributed to the self/the community vs. the city/state (Fig. 6, right, A, B, d).

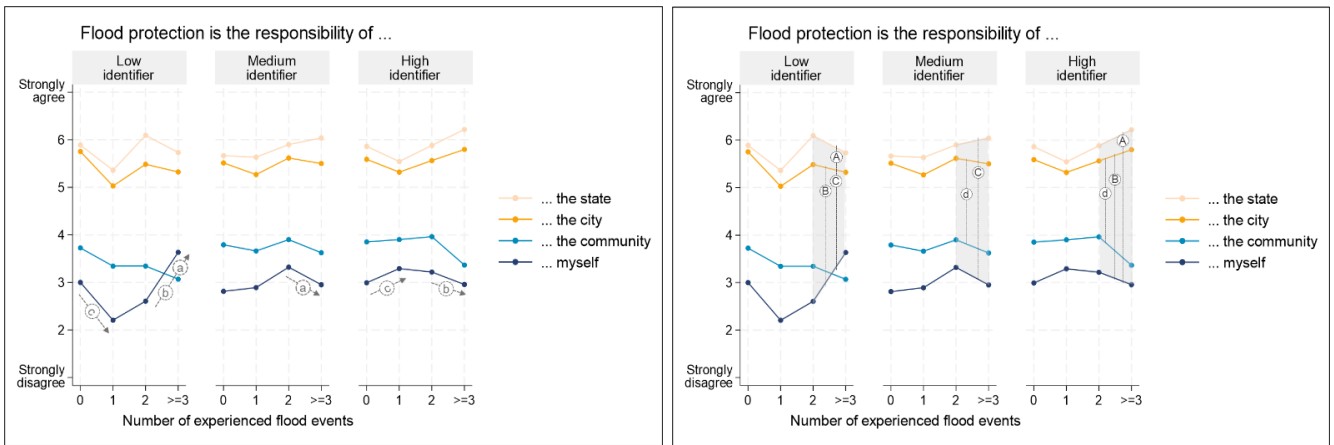

**Figure 6:** Development of ascription of responsibility over flood experience, depending on reported group identification. Left: Statistically significant differences in the effect of flood experience are marked by the same letter, e.g., letter b: the decrease in the personally felt responsibility after the third flood event in the high effective group differs statistically significantly from

the increase in the low identified group. **Right: Statistically significant differences in the diverging development of two responsibility variables are marked with the same letter, e.g., letter A: the increasing difference between the ascribed responsibility to the state and the individual after the third flood event in the high effective group differs statistically significantly from the respective development in the low identified group. Lowercase letter: p<0.05, capitalized letter: p<0.01**

*Perception of social norms*

**Self-efficacy beliefs**

The gap between perceived social versus personal norms increases over flood experience among people with low self-efficacy beliefs (Fig. 7). The augmentation differs from the decrease among respondents who reported medium self-efficacy (Fig. 7, right, a). After the third flood event, the perception of injunctive and descriptive norms increases sharply, accompanied by a decrease in reported moral obligation. The increase in the perception of descriptive norms after the third flood event differs statistically significantly from the development in the other groups (Fig. 7, left, a,b).

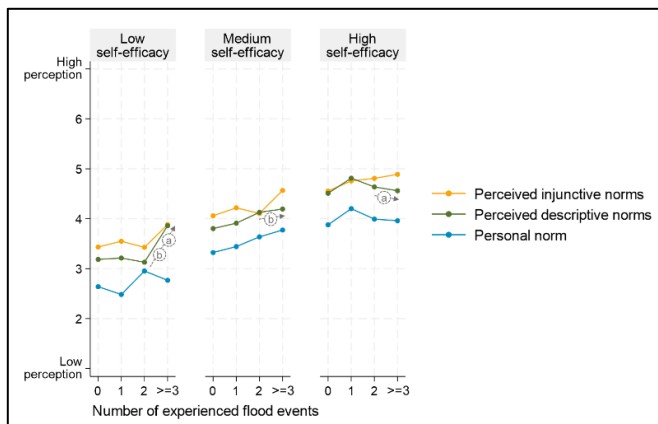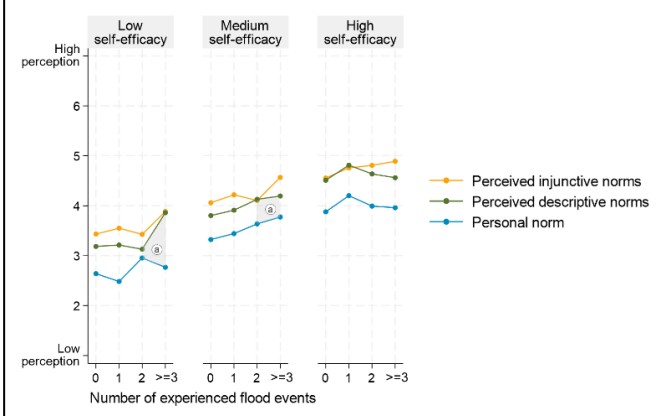

**Figure 7:** Development of norm perception over flood experience, depending on self-efficacy beliefs. **Left: Statistically significant differences in the effect of flood experience are marked by the same letter, e.g., letter a: the decrease in the perceived descriptive norm after the third flood event in the highly effective group differs statistically significantly from the increase in the low-effective group. Right: Statistically significant differences in the diverging development of two norm constructs are marked with the same letter, e.g., a: the increasing difference between reported personal norm and perceived descriptive norm after the third flood event in the low effective group differs statistically significantly from the respective development in the medium effective group. Lowercase letter: p<0.05, capitalized letter: p<0.01; applying heteroskedasticity-consistent standard errors: left: a becomes insignificant (HC3)**

**Strenght of ingroup identification**

We cannot observe a diverging trend between perceived social and personal norms. However, the injunctive norm perception of low identifiers develops statistically significantly differently from the other two groups, supporting the observed pattern under RQ1 (Fig. 8, left, a, b). The increasing gap between the perception of injunctive and personal norms among the low identifiers after the second flood differs significantly from the opposing developments among respondents who reported medium and high ingroup identification (Fig. 8, right, a, b).

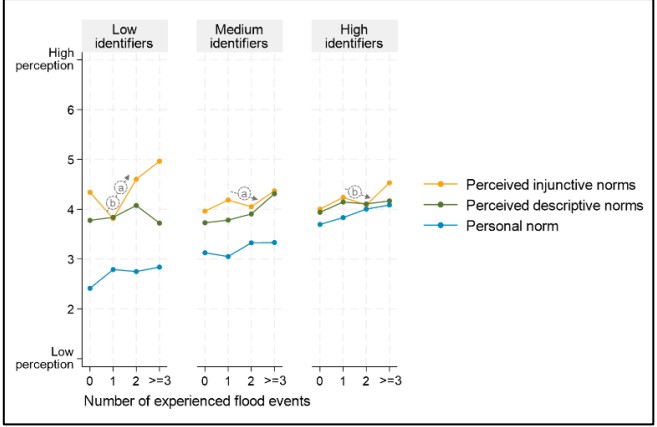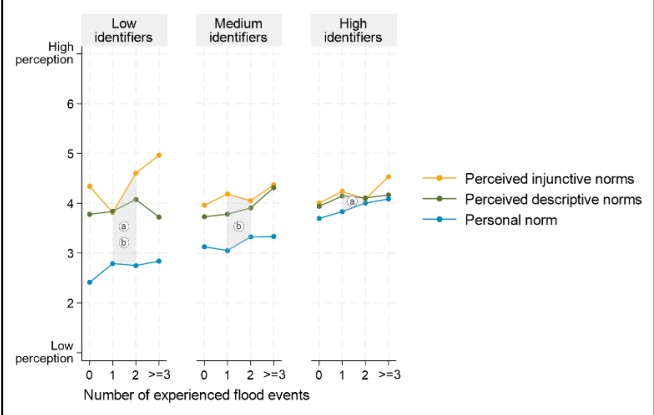

**Figure 8:** Development of norm perception over flood experience, depending on reported group identification. Left: Statistically significant differences in the effect of flood experience are marked by the same letter, e.g., letter b: the increase in the perceived injunctive norm after the second flood event in the low-identified group differs statistically significantly from the decrease in the high-identified group. Right: Statistically significant differences in the diverging development of two norm constructs are marked with the same letter, e.g., letter b: the increasing difference between reported personal norm and perceived injunctive norm after the second flood event in the low identified group differs statistically significantly from the respective development in the medium identified group. Lowercase letter: p<0.05, capitalized letter: p<0.01; applying heteroskedasticity-consistent standard errors: left: a and b become insignificant (HC3), right: a, b become insignificant (HC1)

**Felt helplessness concerning future floods**

The increasing gap between the perception of social and personal norms after the third flood event among respondents who associate either low or high helplessness with future floods differs significantly from the decrease observed in the medium group (Fig. 9, right, a, b). However, we do not find statistically significant differences in the effect of the third flood event (Fig. 9, left).

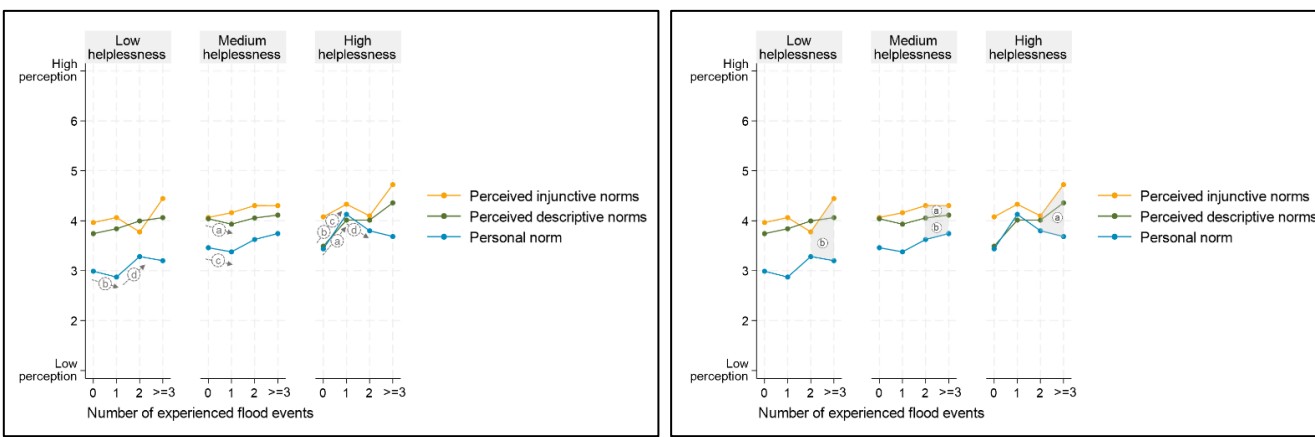

**Figure 9:** Development of norm perception over flood experience, depending on sensed helplessness with regard to future floods. Left: Statistically significant differences in the effect of flood experience are marked by the same letter, e.g., letter d: the increase in the reported personal norm after the second flood event in the low helpless group differs statistically significantly from the decrease in the high helpless group. Right: Statistically significant differences in the diverging development of two norm constructs are marked with the same letter, e.g., letter b: the increasing difference between reported personal norm and perceived injunctive norm after the third flood event in the low helpless group differs statistically significantly from the respective development in the medium helpless group. Lowercase letter: p<0.05, capitalized letter: p<0.01; applying heteroskedasticity-consistent standard errors: right: a, b become insignificant (HC1)

### 5.3 Progressive divergence of ascribed responsibility and perception of norms (RQ3)

The previous analyses reveal an upward trend in the perceived norms for private flood protection over flood experience. At the same time, the responsibility assigned to the community decreases after the third flood, accompanied by an increase in the responsibility attributed to the public authorities. The attribution of responsibility to the self remains widely stable. Testing whether these diverging developments are statistically significant confesses that the increase in the perceived injunctive norm after the third flood event differs significantly from the slight decrease in the attributed responsibility to the self (Fig. 10). The same counts regarding the increasing difference between descriptive norm perception and responsibility attributed to the community (Fig. 11). The remaining combinations of comparable norm perception and responsibility attribution did not show statistically significant diverging developments.

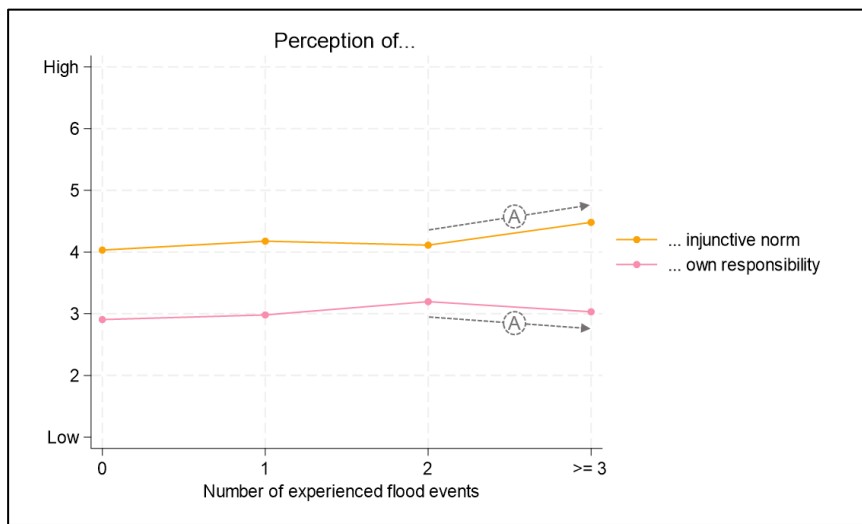

**Figure 10:** Development of injunctive norm perception and personally felt responsibility; A) indicates a statistically significant different effect of the third flood event; capitalized letter: p<0.01

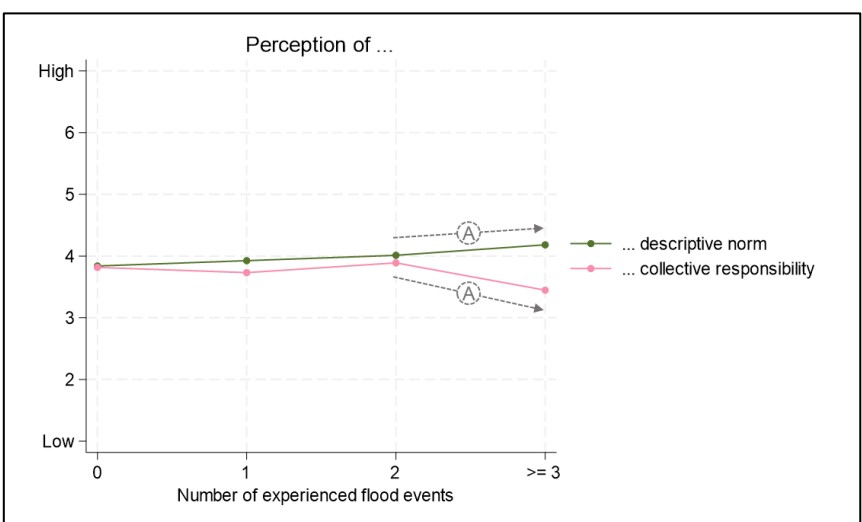

**Figure 11:** Development of descriptive norm perception and reported collective responsibility; A) indicates a statistically significant different effect of the third flood event; lowercase letter: p<0.05, capitalized letter: p<0.01

## 6 Discussion

*I, we, and them: dynamics in attribution of responsibility over flood experience*

We observe a change in the attribution of responsibility among people who have experienced multiple flood events. Respondents who have experienced three or more flood events perceive their local ingroup as less responsible for flood protection, compared to respondents who have little or no flood experience. In contrast, the perceived responsibility of public actors in FRM, such as the city or the state, increases after the second flood event. One explanation could lie in the perceived efficacy of measures that can be taken at the individual, collective, and institutional level. Ueberham et al. (2014) find that people perceive the latter's efficacy as the greatest. Especially for people who have experienced multiple flood events, the efficacy of measures could be a crucial factor when asserting responsibility. Further, Köhler and Han (2024) observe that more experienced people are more likely to associate greater effort with implementing them, which could also be linked to the shift in responsibility attribution.

We identify two groups with an increasing gap between responsibility attributed to the self/the community vs. public authorities: respondents with high self-efficacy beliefs and those highly identified with their local ingroup. Within both groups, the responsibility ascribed to public authorities further increases after the third flood experience, accompanied by a sharp decrease in the ascription to the self/ the community.

Among individuals with high self-efficacy beliefs, motivated cognition may explain their tendency to externalize responsibility when faced with repeated flooding. Motivated cognition is often linked to reducing cognitive dissonance (Frey, 1984, p. 245; 1997, p. 58). Among the respondents who reported high self-efficacy to protect from flooding, this dissonance may arise from the conflict between being affected by floods frequently despite believing in their ability to protect. One way to align both could be to question their self-efficacy. Another way, which we might see in the data, could be to externalize responsibility to explain why efforts may not be as sufficient as intended (see also Cologna et al., 2017; Gosling et al., 2006). Furthermore, the development could be linked to facing limits of private and collective adaptation, a point at which distorting consequences cannot be avoided anymore with the available resources (e.g., Adger et al., 2009; Dow et al., 2013). From their literature review, Kuhlicke et al. (2020b) conclude that the increasing demand for individual- and collective-level adaptation to floods is often not accompanied by sufficient resources. Ueberham et al. (2014) find that more resilient people are more likely to criticize a lack of information on private precautions, hypothesizing that these critical views could stem from them being more aware of existing shortcomings.

For respondents with a high level of group identification, sustaining their self-image could play a role in the externalization of responsibility from "we" to "they". According to the Social Identity Theory (SIT) derive individuals a part of their positive self-regard from their perceptions of their social ingroup, particularly when they assign positive characteristics to it (i.e. their social identities; Tajfel & Turner, 1986). Similarly, groups that are perceived as (highly) agentic can provide a sense of control and being effective (Fritsche, 2022). Explaining multiple affectedness by external responsibility may help to preserve that group's image of being effective, which may, in turn, positively influence self-esteem. Existing research shows that social identities are particularly important for individuals who are highly identified with their group (e.g., Fritsche et al., 2013; Masson et al., 2016; Masson & Fritsche, 2014).

*Shaping of norms: the relationship between frequent flood experience, perception of social norms, and moral obligations*

We find that flood-experienced people are more likely to believe that others expect them to undertake protective measures, implement measures privately, and feel morally obliged to participate in collective actions. This development follows a rather linear path when not accounting for moderating effects.

The observed upward shift in perceived social norms aligns with Fischer & Wiswede (2014, p. 607), stating that events are likely to make them visible, exemplarily through observing the consequences of others' behaviors (Cialdini et al., 1990). We assume that FFE may manifest social norms exemplarily by making the negative consequences of not having adapted visible (e.g., damages at the building level). Additionally, the increased perception of social norms could stem from social interaction, exemplary during reconstruction after a flood. Existing studies find that social interactions can increase people's awareness of others' expectations and behaviors (e.g., Hogg & Reid, 2006; Mackie et al., 2014; Zhang et al., 2023). The upward shift in the moral obligation to participate in collective actions could go in the direction of Fischer & Wiswede's (2014, p. 607) statement that norms can be activated by remembering having obtained comparable help in the past. Another reason could lie in a stronger perception of common fate when undergoing multiple flood events (see also Ntontis et al., 2018).

The sharp increase in the perception of injunctive norms after having experienced at least three floods stands out. Among the tested moderators, we find this pattern for people with low ingroup identification, for whom we also observe an increase in responsibility attributed to the self. There could be various potential explanations for these results. The increase in the perception of injunctive norms could act as a channel to justify feeling personally responsible. Further, the diverging development of descriptive and injunctive norms could hint at a "perception-expectation gap" (perception=descriptive norm, expectation=injunctive norm), potentially reflecting an increasing disconnection from the community. Assumingly, people with low ingroup identification may rather perceive such discrepancy due to sensed uneven distribution of support. However, the increase in personal obligation to participate in collective actions challenges this hypothesis.

We further observe a divergent development between perceived social and personal norms after the third flood event. This is the case for people who reported low self-efficacy beliefs (descriptive vs. personal norm) and great helplessness regarding future floods (injunctive vs. personal norm). Specifically, in both groups, a decrease in reported moral obligation to participate in collective protection actions is accompanied by an increase in the perceived descriptive/injunctive norm. This could be linked to attributing positive characteristics to the ingroup (i.e., ingroup bias) as a channel to restore control when facing a threat to it (e.g., Fritsche et al., 2013; Stollberg et al., 2017). For the current study, this could imply that expectations of others (i.e., injunctive norms) act as a channel to restore control for helpless people, whereas it is others' agency (i.e., descriptive norm) for respondents with low self-efficacy beliefs. Consequently, the combination of low self-efficacy, expected helplessness, and multiple flood experience may lead to an increasing perception of social norms to compensate and find motivation for own protective behavior. The opposite pole is the decline in the reported moral obligation to participate in collective actions. The Norm Activation Theory by Schwartz (1977) postulates that for a moral obligation to develop, individuals must think that they can sufficiently contribute to enhancing the situation. However, low perceived self-efficacy and high helplessness may diminish people's perceived ability and, as a consequence, diminish their moral obligation to contribute to collective flood-protective actions. Our findings indicate that this relationship might be even stronger when experiencing multiple floods.

*Polarization in flood risk management*

The title of this article poses the question whether the experience of multiple flood events leads to a polarization in flood risk management. The answer to this question: it depends.

The increasing gap between the attributed responsibility to the self/the community vs. the city/the state when people have experienced at least three floods hints at a polarization. However, due to the widely stable development of responsibility assigned to the self, the widening gap rather stems from the increased attribution to public authorities, thereby not resulting from divergent developments, in other words, a polarization. Nevertheless, the simultaneous decrease in the responsibility assigned to the community could hint at a polarization of "we" vs. "them".

Moderation analyses reveal that the polarization in responsibility attribution is particularly predominant for certain groups of respondents: those with high self-efficacy beliefs and strong ingroup identification. Furthermore, the analyses detect groups that also show a polarization in the development of perceived norms after the third flood experience: respondents with low self-efficacy beliefs and greater associated helplessness with future floods. Specifically, they are more likely to think that people in their community undertake protective measures privately and also expect the respondent to do so. At the same time, respondents of these groups tend to feel morally less obliged to participate in collective protection actions. For both developments, we hypothesize that maintaining/restoring a positive self-image or a sense of control could be reasons for the observed polarizations. Illustratively, externalizing responsibility to public authorities instead to the community could be a way of justifying why collective protection measures may not be as sufficient as expected without questioning the general efficacy of the community. Particularly for people who highly identify with their community, their self-image of being able to protect from flooding could be closely related to how they perceive their community in this regard. The increase in perceived social norms could be another channel of restoring the own sense of control through the ingroup (see Social Identity Theory, Tajfel & Turner, 1986). Accordingly, assigning agentic characteristics to the community, such as a general consensus on the importance of individual flood protection, might contribute to perceiving the group as capable.

In conclusion, we observe patterns of polarization with respect to the attribution of responsibility and the perception of norms when people experience multiple flood events, particularly pronounced in certain groups. Based on this, we hypothesize that factors related to perceived control over future flood risk and the desire to maintain a capable self-image are drivers of the observed polarization.

From the current study, we derive three implications for disaster risk reduction:

First, we observe that more flood-experienced respondents are less likely to attribute responsibility for flood protection to their community. Here, facing limits of collective community actions might play a role, as also discussed in Kuhlicke et al. (2020b) and Thaler & Seebauer (2019). People who have undergone multiple flood events might be more likely to have been confronted with such limits. Financial support and dissemination of knowledge targeted to region-specific demands may shift these limits

upward. Second, the decreasing personal norm to participate in collective actions to reduce flood risk among respondents with low self-efficacy and control beliefs could be targeted by communicating low-barrier collective actions that do not involve a lot of resources. These could equip individuals with the perception of being able to contribute sufficiently. Third, we linked the higher perception of social norms for private flood protection among respondents with low self-efficacy beliefs and increased feelings of helplessness with regard to future floods to their attempt to restore a sense of control through secondary

control. Perceptions of secondary control could presumably be enhanced through formats like interviews in local newspapers, in which residents share their protective strategies. Additionally, such formats could contribute to normalizing protective behaviors. Importantly, these implications of the current study must be seen against the backdrop of the perception of social norms and responsibility attributions being most likely influenced by the social environment of the respondents, thus limiting the generalization of the results.

*Limitations*

With our study being rather of an exploratory nature, several limitations should be considered when interpreting the results and building upon them.

Only integrating flood experience in the simple regression models ignores other factors that might be related to norm and responsibility perception as well as flood experience. By not accounting for potential confounders, we might be overestimating

the effect of FFE. Generally, the correlational nature of the analyses limits the ability to infer causal relationships.

Further, the group sizes among the moderators partially vary, with some groups being notably small. These small group sizes could increase the sensitivity of the observed relationships to outliers, potentially skewing the distributions of norm perception and responsibility attribution over flood experience.

In addition, the variable measuring participants' perceived ingroup identification was assessed at the city-level. However, this

scale might be too broad to accurately capture the sense of ingroup identification, as is conceptualized in the theories discussed.

Moreover, the statistical significance of the effect of the "third" flood event might be overestimated in some cases, as this group also includes respondents who experienced more than three floods. However, the uncertainty around the measurement does not undermine the observed upward or downward trends.

Lastly, the nonlinear relationships could be linked to specific characteristics of distinct flood events (e.g., variations in

severity). However, whereas we cannot rule out that such bias exists on the individual level, we do not assume that it does play a crucial role in the estimations on the whole sample. The reason for this assumption is that there are variations among the respondents regarding which flood event was their first, second, or third. For example, whereas for some respondents, the 2002 flood event was the first they were affected by, for others it might already have been the second event.

*Directions for future research*

Future research should include replication studies to determine whether the observed dynamics also apply to other social contexts. Moreover, due to the limited existing research, the interpretations of the increasing gaps between perceived norms

and the attribution of responsibility remained speculative. Given that both factors play a crucial role in protective behavior, future studies should focus on the drivers behind this growing discrepancy. Linking other characteristics of the social setting, such as collective efficacy, social cohesion, and interaction to FFE could provide further insight into how changing hazard contexts affect communities. Moreover, our understanding of the link between FFE and the social environment could be expanded by adding other definitions of experience (e.g., vicarious experience, damage level). Since norm perception and responsibility attribution likely depend on contextual factors related to respondents' residences, another avenue for future research could be to apply multilevel analyses that incorporate meso- and macro-level characteristics.

## 7 Conclusion

With the current study, we want to contribute to the discussion around the development of perceived norms and responsibility attribution in FRM. Here, we examine the role played by frequent flood experience. Against the backdrop of global climate change causing more frequent flooding in many parts of the world, individual-level protective behavior being a fundamental pillar in reducing flood risk, and social norms and responsibility attribution playing a pivotal role in motivating protective behavior, it is vital to understand how these factors interrelate.

So far, there is only little knowledge on the sensitivity of responsibility attribution and norm perception to flood experience in the scholarly literature. Therefore, we began by exploratively identifying respective patterns in the data and interpreted the findings based on sociopsychological theories and concepts. From this we hypothesize that especially social learning and maintaining a positive image of the self through the ingroup could play a role in the detected relationships. We additionally observe a polarization in responsibility attribution to the community vs. to public authorities. Moderation analyses suggest that the polarization is particularly pronounced for people with high self-efficacy beliefs regarding individual flood protection and strong ingroup identification. For people with low self-efficacy beliefs and low expected controllability of future floods, a polarization between an increase in perceived social norms versus a decrease in moral obligation to participate in collective actions can be observed. From these results, we conclude that perceived control over future flood risk and ingroup identification could have a polarizing effect.

In summary, with the current study, we shed light on a potential sensitivity of norm perception and responsibility attribution to frequent flood experience. Due to the exploratory nature of the study under the application of a single dataset, we strongly encourage fellow researchers to replicate the current study to test whether the patterns found are also observable for other social contexts. Furthermore, considering additional moderators in future studies may even enhance the understanding of the observed relationships in the current study.

*Acknowledgments:* The data was collected in the context of the PIVO project, which was financially supported by the German Federal Ministry for the Environment, Nature Conservation, Nuclear Safety and Consumer Protection (FKZ 3718 48 101 0). The authors would like to thank Sabrina Köhler and Sebastian Bamberg for conceptualizing the survey and Daniela Siedschlag and many student assistants for conducting the survey.

*Data and code availability:* Data used for this study and codes for the analyses can be made available for scientific use upon request to the author.

*Author contributions:* LK, TM, SH and CK conceptualized this study. LK and TM developed the model. LK performed the formal analysis and prepared the results. The original draft was written by LK and reviewed and edited by TM, SH, and CK. LK prepared the final manuscript for submission.

*Competing interests:* The authors declare that they have no conflict of interest.

*Financial support:* This research has been supported by the Deutsche Forschungsgemeinschaft (grant no. GRK 2043, project number 25103684) as well as by the Federal Ministry for the Environment, Nature Conservation, Nuclear Safety and Consumer Protection/German Environmental Office through the PIVO project (UBA, Forschungskennzahl 3718 48 101 0).

During the preparation of this work the authors used Writefull in order to improve the language. After using these tools/services, the authors reviewed and edited the content as needed and takefull responsibility for the content of the publication.

## Appendix A: Relationship between variables

Table A1: Correlation matrix of the included variables

| Spearman Correlations | Descr.norm | Pers. norm | Pers. resp. | Coll. resp. | City's resp. | State's resp. | FFE | Self-efficacy | Group ident. | Helplessness |
|---|---|---|---|---|---|---|---|---|---|---|
| Inj. norm | 0.44** | 0.1** | 0.19** | 0.16** | 0.12** | 0.05* | 0.07** | 0.23** | -0.01 | 0.07** |
| Descr. norm | | 0.16** | 0.17** | 0.17** | 0.08** | 0.002 | 0.08** | 0.28** | 0.06* | 0.03 |
| Pers. norm | | | 0.19** | 0.32** | 0.005 | 0.01 | 0.07** | 0.25** | 0.23** | 0.14** |
| Pers. resp. | | | | 0.46** | -0.06* | -0.11** | 0.05* | 0.23** | 0.04 | -0.007 |
| Coll. resp. | | | | | 0.07** | -0.02 | -0.04 | 0.16** | 0.03 | 0.03 |
| City's resp. | | | | | | 0.07** | 0.03 | 0.002 | 0.02 | 0.15** |
| State's resp. | | | | | | | 0.07** | -0.004 | 0.03 | 0.12** |
| FFE | | | | | | | | 0.002 | 0.01 | 0.18 |
| Self-efficacy | | | | | | | | | 0.03 | -0.17** |
| Group ident. | | | | | | | | | | 0.1** |

## Appendix B: Regression tables for single interactions

Table B1: Standardized interaction coefficients for responsibility attribution

| Responsibility variable | Flood experience | Interaction | | | | | |
|---|---|---|---|---|---|---|---|
| | | Self-efficacy beliefs | | | Group identification | | |
| | | Low vs. medium | Low vs. high | Medium vs. high | Low vs. medium | Low vs. high | Medium vs. high |
| Personal responsibility | 1 | 0.18 [-0.12;0.48] | 0.21 [-0.17;0.58] | 0.03 [-0.3;0.36] | 0.39 [-0.06;0.83] | 0.51* [0.06;0.96] c | 0.12 [-0.15;0.39] |
| | 2 | -0.1 [-0.45; 0.24] | 0.11 [-0.33;0.54] | 0.21 [-0.17;0.58] | 0.05 [-0.45;0.56] | -0.21 [-0.73;0.31] | -0.26 [-0.57;0.04] |
| | 3 | 0.06 [-0.32;0.44] | -0.56*a [-1.04;-0.08] | -0.62**B [-1.02;-0.22] | -0.68*a [-1.21;-0.14] | -0.63*b [-1.17;-0.09] | 0.05 [-0.29;0.38] |
| | Adj. R-squared | 0.0569 | | | 0.007 | | |
| | N | 1613 | | | 1640 | | |
| | F-statistic | 9.85** | | | 2.09* | | |

| Responsibility variable | Flood experience | Self-efficacy beliefs | | | Group identification | | |
|---|---|---|---|---|---|---|---|
| | | Low vs. medium | Low vs. high | Medium vs. high | Low vs. medium | Low vs. high | Medium vs. high |
| Collective responsibility | 1 | 0.18 [-0.12; 0.49] | 0.45* [f] [0.07; 0.83] | 0.27 [-0.06;0.6] | 0.17 [-0.27;0.6] | 0.3 [-0.15;0.75] | 0.13 [0.13;0.4] |
| | 2 | -0.43* [e] [-0.78; -0.09] | -0.53* [g] [-0.97;0.09] | -0.09 [-0.47;0.28] | 0.19 [-0.31;0.69] | 0.04 [-0.68;0.4] | -0.15 [-0.45;0.16] |
| | 3 | 0.28 [-0.11;0.66] | -0.1 [-0.58; 0.38] | -0.38 [-0.78;0.02] | -0.014 [-0.56;0.53] | -0.18 [-0.73;0.37] | -0.16 [-0.49;0.17] |
| | Adj. R-squared | 0.04 | | | 0.005 | | |
| | N | 1610 | | | 1644 | | |
| | F-statistic | 6.42** | | | 1.8* | | |
| City's responsibility | 1 | 0.012 [-0.29;0.32] | 0.08 [-0.3;0.5] | 0.07 [-0.26;0.4] | 0.41 [0.027;0.84] | 0.39 [-0.05; 0.83] | -0.01 [-0.28;0.25] |
| | 2 | -0.14 [-0.49;0.21] | -0.2 [-0.64;0.25] | -0.05 [-0.43;0.32] | -0.18 [-0.68;0.32] | -0.26 [-0.77;0.25] | -0.08 [-0.39;0.22] |
| | 3 | 0.43* [c] [0.05;0.82] | 0.56* [d] [0.08;1.04] | 0.12 [-0.28;0.53] | 0.03 [-0.5;0.56] | 0.3 [-0.24; 0.84] | 0.27 [-0.06;0.59] |
| | Adj. R-squared | 0.008 | | | 0.006 | | |
| | N | 1635 | | | 1679 | | |
| | F-statistic | 2.19* | | | 1.96* | | |
| State's responsibility | 1 | 0.14 [-0.17;0.44] | -0.0001 [-0.38;0.38] | -0.14 [-0.47;0.2] | 0.42 [-0.01;0.86] | 0.23 [-0.21;0.68] | -0.19 [-0.45;0.08] |
| | 2 | -0.23 [-0.58;0.12] | -0.15 [-0.6;0.3] | 0.08 [-0.3;0.46] | -0.43 [-0.93;0.07] | -0.36 [-0.88;0.15] | 0.07 [-0.23;0.38] |
| | 3 | 0.11 [-0.28;0.49] | 0.32 [-0.17;0.8] | 0.21 [-0.2;0.62] | 0.35 [-0.19;0.88] | 0.46 [-0.09;1] | 0.11 [-0.22;0.44] |
| | Adj. R-squared | 0.01 | | | 0.01 | | |
| | N | 1625 | | | 1666 | | |
| | F-statistic | 2.7** | | | 3.04** | | |

*: p<0.05. **: p<0.01; 95%-CI in square brackets; superscript letters refer to the effects in the graphs of the main analyses; bold p-values are p-values of coefficients that are not significant when applying heteroskedasticity-robust standard errors (HC3)

Table B2: Interaction coefficients and significance, single interactions, responsibility attribution, robust, standardized

| Responsibility variable | Flood experience | Interaction | | | | | |
|---|---|---|---|---|---|---|---|
| | | Self-efficacy beliefs | | | Group identification | | |
| | | Low vs. medium | Low vs. high | Medium vs. high | Low vs. medium | Low vs. high | Medium vs. high |
| Personal responsibility | 1 | 0.18 [-0.09;0.45] | 0.21 [-0.17;0.58] | 0.03 [-0.32;0.38] | 0.39 [-0.07;0.84] | 0.51* [0.04;0.98] | 0.12 [-0.14;0.39] |
| | 2 | -0.1 [-0.42;0.22] | -0.11 [-0.34;0.55] | 0.21 [-0.19;0.61] | 0.05 [-0.42;0.53] | -0.21 [-0.71;0.29] | -0.26 [-0.57;0.05] |
| | 3 | 0.06 [-0.31;0.43] | -0.56* [-1.06;-0.06] | -0.62** [-1.06;-0.18] | -0.68* [-1.24;-0.11] | -0.63* [-1.21;-0.05] | 0.05 [-0.29;0.38] |
| | R-squared | 0.06 | | | 0.01 | | |
| | N | 1613 | | | 1640 | | |
| | F-statistic | 10.4** | | | 2.1** | | |
| Collective responsibility | 1 | 0.18 [-0.12;0.48] | 0.45* [0.07;0.83] | 0.27 [-0.06;0.6] | 0.17 [-0.32;0.65] | 0.3 [-0.21;0.81] | 0.13 [-0.13;0.4] |
| | 2 | -0.43* [-0.8;-0.07] | -0.53* [-0.98;-0.08] | -0.09 [-0.46;0.27] | 0.19 [-0.34;0.71] | 0.04 [-0.51;0.59] | -0.15 [-0.46;0.17] |

| | | Self-efficacy beliefs | | | Group identification | | |
|---|---|---|---|---|---|---|---|
| | 3 | 0.28 [-0.14;0.69] | -0.1 [-0.61;0.41] | -0.38 [-0.79;0.03] | -0.01 [-0.55;0.52] | -0.18 [-0.73;0.38] | -0.16 [-0.51;0.18] |
| | R-squared | 0.04 | | | 0.012 | | |
| | N | 1610 | | | 1644 | | |
| | F-statistic | 6.5** | | | 1.76 | | |
| City's responsibility | 1 | 0.012 [-0.32;0.34] | 0.08 [-0.33;0.5] | 0.07 [-0.27;0.41] | 0.41 [-0.12;0.94] | 0.39 [-0.15;0.93] | -0.01 [-0.28;0.25] |
| | 2 | -0.14 [-0.52;0.23] | -0.2 [-0.67;0.28] | -0.05 [-0.45;0.34] | -0.18 [-0.77;0.41] | -0.26 [0.87;0.34] | -0.08 [-0.39;0.22] |
| | 3 | 0.43 **(p=0.05)** [-0.008;0.88] | 0.56* [0.02;1.1] | 0.13 [-0.28;0.53] | 0.03 [-0.58;0.64] | 0.3 [-0.33;0.92] | 0.27 [-0.06; 0.6] |
| | R-squared | 0.01 | | | 0.01 | | |
| | N | 1635 | | | 1679 | | |
| | F-statistic | 1.97* | | | 1.72 | | |
| State's responsibility | 1 | 0.14 [-0.21;0.48] | -0.0001 [-0.44;0.44] | -0.14 [-0.5;0.23] | 0.42 [-0.14;0.99] | 0.23 [-0.35;0.82] | -0.19 [-0.47;0.09] |
| | 2 | -0.23 [-0.6;0.13] | -0.15 [-0.62;0.32] | 0.08 [-0.31;0.48] | -0.43 [-0.99;0.12] | -0.36 [-0.94;0.22] | 0.07 [-0.24;0.38] |
| | 3 | 0.11 [-0.27;0.48] | 0.32 [-0.14;0.77] | 0.21 [-0.15;0.57] | 0.35 [-0.19;0.88] | 0.46 [-0.09;1] | 0.12 [-0.18;0.4] |
| | R-squared | 0.02 | | | 0.02 | | |
| | N | 1625 | | | 1666 | | |
| | F-statistic | 2.87** | | | 3.28** | | |

*: p<0.05. **: p<0.01; 95%-CI in square brackets; superscript letters refer to the effects in the graphs of the main analyses; bold p-values are p-values of coefficients that are not significant when applying heteroskedasticity-robust standard errors (HC3)

Table B3: Interaction coefficients and significance, single interaction, norm perception, standardized

| Norm variable | Flood experience | Interaction | | | | | |
|---|---|---|---|---|---|---|---|
| | | Self-efficacy beliefs | | | Group identification | | |
| | | Low vs. medium | Low vs. high | Medium vs. high | Low vs. medium | Low vs. high | Medium vs. high |
| Injunctive norm | 1 | 0.02 [-0.29;0.33] | 0.05 [-0.33;0.43 | 0.03 [-0.31;0.36] | 0.43 [-0.01;0.88] | 0.44 [-0.01;0.9] | 0.007 [-0.27;0.28] |
| | 2 | 0.005 [-0.35;0.36] | 0.1 [-0.35;0.55] | 0.09 [-0.29;0.48] | -0.53*[a] [-1.05;-0.02] | -0.54*[b] [-1.07;-0.02] | -0.009 [-0.32;0.3] |
| | 3 | 0.003 [-0.38;0.39] | -0.22 [-0.7;0.27] | -0.22 [-0.63;0.19] | -0.02 [-0.57;0.52] | 0.05 [-0.51;0.6] | 0.07 [-0.26;0.41] |
| | Adj. R-squared | 0.06 | | | 0.009 | | |
| | N | 1584 | | | 1606 | | |
| | F-statistic | 9.68** | | | 2.29** | | |
| Descriptive norm | 1 | 0.05 [-0.25;0.35] | 0.17 [-0.2;0.55] | 0.12 [-0.2;0.45] | -0.004 [-0.46;0.44] | -0.09 [-0.37;0.55] | 0.09 [-0.18;0.37] |
| | 2 | 0.19 [-0.16;0.53] | -0.06 [-0.49;0.38] | -0.24 [-0.61;0.13] | -0.07 [-0.59;0.44] | -0.17 [-0.7;0.36] | -0.1 [-0.41;0.21] |
| | 3 | -0.41*[b] [-0.79;-0.03] | -0.5*[a] [-0.98;-0.02] | -0.09 [-0.49;0.31] | 0.47 [-0.08;1.02] | 0.26 [-0.31;0.82] | -0.21 [-0.55;0.12] |

| | | | | | | | |
|---|---|---|---|---|---|---|---|
| Adj. R-squared | | 0.08 | | | 0.007 | | |
| N | | 1586 | | | 1611 | | |
| F-statistic | | 14.06** | | | 1.99* | | |
| Personal norm | 1 | 0.15 [-0.14;0.45] | 0.26 [-0.1;0.63] | 0.11 [-0.21;0.44] | -0.25 [-0.67;0.17] | -0.13 [-0.56;0.3] | 0.12 [-0.14;0.38] |
| | 2 | -0.15 [-0.49;0.18] | -0.37 [-0.8;0.05] | -0.22 [-0.59;0.15] | 0.17 [-0.32;0.66] | 0.12 [-0.39;0.62] | -0.06 [-0.35;0.24] |
| | 3 | 0.18 [-0.2;0.56] | 0.08 [-0.39;0.56] | -0.09 [-0.49;0.31] | -0.05 [-0.57;0.48] | -0.004 [-0.54;0.53] | 0.04 [-0.29;0.37] |
| Adj. R-squared | | 0.06 | | | 0.05 | | |
| N | | 1653 | | | 1663 | | |
| F-statistic | | 11.28** | | | 8.99** | | |

*: p<0.05. **: p<0.01; 95%-CI in square brackets; superscript letters refer to the effects in the graphs of the main analyses; bold p-values are p-values of coefficients that are not significant when applying heteroskedasticity-robust standard errors (HC3)

| Norm variable | Flood experience | Helplessness | | |
|---|---|---|---|---|
| | | Low vs. medium | Low vs. high | Medium vs. high |
| Injunctive norm | 1 | -0.003 [-0.29;0.28] | 0.09 [-0.28;0.45] | 0.09 [-0.26;0.45] |
| | 2 | 0.25 [-0.08;0.58] | 0.03 [-0.37;0.43] | -0.22 [-0.6;0.16] |
| | 3 | -0.39 [-0.78;0.002] | -0.02 [-0.44;0.4] | 0.36 [-0.01;0.74] |
| Adj. R-squared | | 0.009 | | |
| N | | 1623 | | |
| F-statistic | | 2.38** | | |
| Descriptive norm | 1 | -0.13 [-0.41;0.16] | 0.26 [-0.1;0.62] | 0.4*[a] [0.04;0.74] |
| | 2 | -0.02 [-0.35;0.31] | -0.1 [-0.5;0.3] | -0.08 [-0.45;0.3] |
| | 3 | -0.006 [-0.39;0.38] | 0.17 [-0.25;0.59] | 0.18 [-0.2;0.55] |
| Adj. R-squared | | 0.008 | | |
| N | | 1628 | | |
| F-statistic | | 2.14* | | |
| Personal norm | 1 | 0.02 [-0.25;0.3] | 0.45*[b] [0.1;0.8] | 0.43*[c] [0.09;0.77] |
| | 2 | -0.09 [-0.41;0.23] | -0.41*[d] [-0.79;-0.02] | -0.32 [-0.68;0.05] |
| | 3 | 0.11 [-0.27;0.49] | -0.02 [-0.43;0.39] | -0.13 [-0.5;0.24] |
| Adj. R-squared | | 0.03 | | |
| N | | 1683 | | |
| F-statistic | | 4.96** | | |

*: p<0.05. **: p<0.01; 95%-CI in square brackets; superscript letters refer to the effects in the graphs of the main analyses; bold p-values are p-values of coefficients that are not significant when applying heteroskedasticity-robust standard errors (HC3)

Table B4: Interaction coefficients and significance, single interaction, norm perception, robust, standardized

| Norm variable | FFE | Interaction | | | | | |
|---|---|---|---|---|---|---|---|
| | | Self-efficacy beliefs | | | Group identification | | |
| | | Low vs. medium | Low vs. high | Medium vs. high | Low vs. medium | Low vs. high | Medium vs. high |
| Injunctive norm | 1 | 0.02 [-0.32;0.37] | 0.05 [-0.37;0.47] | 0.03 [-0.3;0.35] | 0.43 [-0.08; 0.95] | 0.44 [-0.08;0.96] | 0.007 [-0.26;0.27] |
| | 2 | 0.005 [-0.38;0.39] | 0.1 [-0.38;0.58] | 0.09 [-0.27;0.46] | -0.53 (p=0.08) [-1.15;0.08] | -0.54 (p=0.08) [-1.17;0.08] | -0.009 [-0.3;0.28] |
| | 3 | 0.003 [-0.43;0.43] | -0.22 [-0.74; 0.3] | -0.22 [-0.62;0.18] | -0.02 [-0.66;0.61] | 0.05 [-0.58;0.67] | 0.07 [-0.25;0.4] |
| | R-squared | 0.06 | | | 0.02 | | |
| | N | 1584 | | | 1606 | | |
| | F-statistic | 8.49** | | | 2.13* | | |
| Descriptive norm | 1 | 0.05 [-0.27; 0.37] | 0.17 [-0.23;0.57] | 0.12 [-0.22;0.46] | -0.004 [-0.55;0.54] | 0.09 [-0.47; 0.65] | 0.09 [-0.18; 0.36] |
| | 2 | 0.19 [-0.18;0.55] | -0.06 [-0.52;0.41] | -0.24 [-0.61; 0.13] | -0.07 [-0.66;0.52] | -0.17 [-0.78; 0.44] | -0.1 [-0.4;0.21] |
| | 3 | -0.41* [-0.82;-0.002] | -0.5 (p=0.06) [-1.02; 0.02] | -0.09 [-0.5;0.32] | 0.47 [-0.07; 1.01] | 0.26 [-0.3;0.81] | -0.21 [0.54;0.11] |
| | R-squared | 0.09 | | | 0.01 | | |
| | N | 1586 | | | 1611 | | |
| | F-statistic | 12.84** | | | 2.04* | | |
| Personal norm | 1 | 0.15 [-0.13;0.43] | 0.26 [-0.11;0.64] | 0.11 [-0.23;0.46] | -0.25 [-0.69; 0.19] | -0.13 [-0.59; 0.32] | 0.12 [-0.15;0.38] |
| | 2 | -0.15 [-0.48;0.18] | -0.37 [-0.82;0.07] | -0.22 [-0.61; 0.16] | 0.17 [-0.35;0.7] | 0.12 [-0.43;0.66] | -0.06 [-0.35;0.23] |
| | 3 | 0.18 [-0.22;0.58] | 0.08 [-0.43;0.6] | -0.1 [-0.52;0.33] | -0.05 [-0.63;0.54] | -0.004 [-0.61;0.6] | 0.04 [-0.29;0.37] |
| | R-squared | 0.07 | | | 0.06 | | |
| | N | 1653 | | | 1663 | | |
| | F-statistic | 11.4** | | | 8.92** | | |

*: p<0.05. **: p<0.01; 95%-CI in square brackets; superscript letters refer to the effects in the graphs of the main analyses; bold p-values are p-values of coefficients that are not significant when applying heteroskedasticity-robust standard errors (HC3)

| Norm variable | Flood experience | | | |
|---|---|---|---|---|
| | | Helplessness | | |
| | | Low vs. medium | Low vs. high | Medium vs. high |
| Injunctive norm | 1 | -0.003 [-0.29;0.29] | 0.09 [-0.3;0.48] | 0.09 [-0.27;0.45] |

| | | | | |
|---|---|---|---|---|
| | 2 | 0.25 [-0.07;0.57] | 0.03 [-0.4;0.47] | -0.22 [-0.61;0.18] |
| | 3 | -0.39* [-0.75;-0.02] | -0.02 [-0.46;0.41] | [- 0.36 [-0.03;0.76] |
| | R-squared | 0.02 | | |
| | N | 1623 | | |
| | F-statistic | 2.25* | | |
| Descriptive norm | 1 | -0.13 [-0.41;0.16] | 0.26 [-0.15;0.67] | 0.39* [0.007; 0.77] |
| | 2 | -0.02 [-0.35;0.31] | -0.1 [-0.53;0.34] | -0.08 [-0.47;0.32] |
| | 3 | -0.006 [-0.39;0.38] | 0.17 [-0.25;0.6] | 0.18 [-0.19;0.55] |
| | R-squared | 0.01 | | |
| | N | 1628 | | |
| | F-statistic | 2* | | |
| Personal norm | 1 | 0.02 [-0.25;0.28] | 0.45* [0.06;0.83] | 0.43* [0.06;0.8] |
| | 2 | -0.09 [-0.4;0.22] | -0.41* [-0.81;-0.01] | -0.32 [-0.7;0.06] |
| | 3 | 0.11 [-0.28;0.5] | -0.02 [-0.5;0.43] | -0.13 [-0.51;0.26] |
| | R-squared | 0.03 | | |
| | N | 1683 | | |
| | F-statistic | 4.77** | | |

*: $p<0.05$. **: $p<0.01$; 95%-CI in square brackets; superscript letters refer to the effects in the graphs of the main analyses; bold p-values are p-values of coefficients that are not significant when applying heteroskedasticity-robust standard errors (HC3)Appendix C: Regression tables for double interactions

**Appendix C: Regression tables for double interactions**

Table C1: Interaction coefficients and significance, double interactions, responsibility attribution; 95%-CI in brackets, standardized

| Responsibility variable | | FFE | Interaction | | | | | |
|---|---|---|---|---|---|---|---|---|
| | | | Self-efficacy beliefs | | | Group identification | | |
| | | | Low vs. medium | Low vs. high | Medium vs. high | Low vs. medium | Low vs. high | Medium vs. high |
| Personal resp. vs. | Coll. resp. | 2 → 3 | 0.22 [-0.27;0.71] | 0.45 [-0.16;1.07] | 0.23 [-0.29;0.74] | 0.64 [-0.03;1.31] | 0.41 [-0.27;1.1] | -0.23 [-0.64;0.19] |
| | City's resp. | 2 → 3 | 0.38 [-0.11;0.86] | 1.1**E [0.49;1.71] | 0.73**A [0.22;1.24] | 0.65 [-0.01;1.32] | 0.87*B [0.2;1.55] | 0.22 [-0.19;0.63] |
| | State's resp. | 2 → 3 | 0.05 [-0.44;0.54] | 0.87**D [0.25;1.48] | 0.82**B [0.3;1.33] | 0.95**C [0.27;1.62] | 1.01**A [0.33;1.7] | 0.07 [-0.35;0.48] |
| Collective resp. vs. | City's resp. | 2 → 3 | 0.15 [-0.34;0.64] | 0.65*f [0.04;1.26] | 0.5 [-0.01;1] | 0.02 [-0.65;0.69] | 0.46 [-0.22;1.15] | 0.45*d [0.03;0.86] |
| | State's resp. | 2 → 3 | -0.18 [-0.66;0.31] | 0.41 [-0.2;1.02] | 0.59*C [0.08;1.1] | 0.31 [-0.37;0.98] | 0.6 [-0.09;1.29] | 0.29 [-0.12;0.71] |
| City's resp. | State's resp. | 2 → 3 | -0.33 [-0.81; 0.16] | -0.24 [-0.84;0.37] | 0.09 [-0.42;0.6] | 0.29 [-0.38;0.96] | 0.14 [-0.54;0.82] | -0.15 [-0.57;0.26] |
| N | | | 1648 | | | 1698 | | |

Table C2: Interaction coefficients and significance, double interactions, responsibility attribution; 95%-CI in brackets, robust, standardized

| Responsibility variable | | FFE | Interaction | | | | | |
|---|---|---|---|---|---|---|---|---|
| | | | Self-efficacy beliefs | | | Group identification | | |
| | | | Low vs. medium | Low vs. high | Medium vs. high | Low vs. medium | Low vs. high | Medium vs. high |
| Personal resp. vs. | Coll. resp. | 2 → 3 | 0.22 [-0.2;0.65] | 0.45 [-0.09;1] | 0.23 [-0.21;0.67] | 0.64 [-0.001;1.28] | 0.41 [-0.23;1.06] | -0.23 [-0.55;0.1] |
| | City's resp. | 2 → 3 | 0.38 [-0.2;0.95] | 1.1** [0.36;1.84] | 0.73* [0.13;1.33] | 0.65 [-0.17;1.48] | 0.87* [0.04;1.71] | 0.22 [-0.24;0.68] |
| | State's resp. | 2 → 3 | 0.05 [-0.47;0.57] | 0.87* [0.17;1.56] | 0.82** [0.22;1.42] | 0.95* [0.17;1.71] | 1.01* [0.23;1.79] | 0.07 [-0.39;0.52] |
| Collective resp. vs. | City's resp. | 2 → 3 | 0.15 [-0.43;0.73] | 0.65 (p=0.08) [-0.07;1.37] | 0.5 [-0.07;1.07] | 0.02 [-0.74;0.77] | 0.46 [-0.31;1.23] | 0.45 [-0.003;0.89] |
| | State's resp. | 2 → 3 | -0.18 [-0.7;0.35] | 0.41 [-0.26;1.09] | 0.59* [0.03;1.15] | 0.31 [-0.42;1.03] | 0.6 [-0.14;1.34] | 0.29 [-0.15;0.74] |
| City's resp. | State's resp. | 2 → 3 | -0.33 [-0.71;0.06] | -0.24 [-0.69;0.22] | 0.09 [-0.26;0.44] | 0.29 [-0.19;0.77] | 0.14 [-0.34;0.62] | -0.15 [-0.45;0.14] |
| N | | | 1648 | | | 1698 | | |

*: p<0.05. **: p<0.01; 95%-CI in square brackets; superscript letters refer to the effects in the graphs of the main analyses; bold p-values are p-values of coefficients that are not significant when applying heteroskedasticity-robust

standard errors (HC1)

Table C3: Interaction coefficients and significance, double interaction, norm perception; 95%-CI in brackets, standardized

| Norm variable | | FFE | Interaction | | | | | |
|---|---|---|---|---|---|---|---|---|
| | | | Self-efficacy beliefs | | | Group identification | | |
| | | | Low vs. medium | Low vs. high | Medium vs. high | Low vs. medium | Low vs. high | Medium vs. high |
| Inductive norm vs. | Descriptive norm | 2 → 3 | -0.4 [-0.89;0.08] | -0.26 [-0.87;0.34] | 0.14 [-0.37;0.65] | 0.47 [-0.2;1.14] | 0.19 [-0.49;0.87] | -0.28 [-0.69;0.12] |
| | | 1 → 2 | | | | 0.44 [-0.19;1.06] | 0.35 [-0.29;0.99] | -0.09 [-0.46;0.29] |
| | Personal norm | 2 → 3 | 0.18 [-0.2;1.66] | 0.31 [-0.26;0.92] | 0.13 [-0.38;0.64] | -0.03 [-0.69;0.62] | -0.1 [-0.77;0.57] | -0.07 [-0.47;0.34] |
| | | 1 → 2 | | | | 0.73*[b] [0.12;1.35] | 0.69*[a] [0.05;1.32] | -0.04 [-0.42;0.33] |
| Descriptive norm | Personal norm | 2 → 3 | 0.58*[a] [0.1;1.07] | 0.57 [-0.04;1.18] | -0.01 [-0.52;0.5] | -0.51 [-1.17;0.16] | -0.29 [-0.96;0.39] | 0.22 [-0.19;1.62] |
| | | 1 → 2 | | | | 0.3 [-0.32;0.92] | 0.34 [-0.3;0.98] | 0.04 [-0.32;0.42] |
| N | | | 1681 | | | 1712 | | |

| Norm variable | | FFE | Interaction | | |
|---|---|---|---|---|---|
| | | | Helplessness | | |
| | | | Low vs. medium | Low vs. high | Medium vs. high |
| Injunctive norm vs. | Descriptive norm | 2 → 3 | 0.39 [-0.08;0.87] | 0.19 [-0.32;0.71] | -0.2 [-0.66;0.26] |
| | Personal norm | 2 → 3 | 0.49*[b] [0.01;0.96] | -0.02 [-0.53;0.49] | -0.51*[a][-0.97;-0.05] |
| Descriptive norm | Personal norm | 2 → 3 | 0.09 [-0.38;0.56] | -0.21 [-0.73;0.3] | -0.31 [-0.77;0.15] |
| N | | | 1731 | | |

Table C4: Interaction coefficients and significance, double interaction, norm perception; 95%-CI in brackets, robust, standardized

| Norm variable | | FFE | Interaction | | | | | |
|---|---|---|---|---|---|---|---|---|
| | | | Self-efficacy beliefs | | | Group identification | | |
| | | | Low vs. medium | Low vs. high | Medium vs. high | Low vs. medium | Low vs. high | Medium vs. high |
| Inductive norm vs. | Descriptive norm | 2 → 3 | -0.4 [-0.82;0.01] | -0.26 [-0.78;0.26] | 0.14 [-0.3;0.56] | 0.47 [-0.2;1.14] | 0.19 [-0.49;0.86] | -0.28 [-0.62;0.05] |
| | | 1 → 2 | | | | 0.44 [-0.17;1.05] | 0.35 [-0.27;0.97] | -0.09 [-0.4;0.22] |
| | Personal norm | 2 → 3 | 0.18 [-0.34;0.71] | 0.31 [-0.38;1] | 0.13 [-0.46;0.72] | -0.03 [-0.86;0.79] | -0.1 [-0.94;0.74] | -0.07 [-0.5;0.37] |
| | | 1 → 2 | | | | 0.73 (p=0.08) [-0.08;1.55] | 0.69 (p=0.099) [-0.13;1.51] | -0.04 [-0.42;0.33] |
| Descriptive norm | Personal norm | 2 → 3 | 0.58* [0.06;1.11] | 0.57 [-0.12;1.27] | -0.011 [-0.6;0.57] | -0.51 [-1.3;0.29] | -0.29 [-1.1;0.53] | 0.22 [-0.22;0.66] |
| | | 1 → 2 | | | | 0.3 [-0.46;1.05] | 0.34 [-0.44;1.11] | 0.04 [-0.33;0.42] |
| N | | | 1681 | | | 1712 | | |

| Norm variable | | FFE | Interaction | | |
|---|---|---|---|---|---|
| | | | Helplessness | | |
| | | | Low vs. medium | Low vs. high | Medium vs. high |
| Inductive norm vs. | Descriptive norm | 2 → 3 | 0.39* [0.01;0.78] | 0.19 [-0.27;0.65] | -0.2 [-0.62;0.21] |

| | Personal norm | 2 → 3 | 0.49 **(p=0.07)** [-0.05;1.02] | -0.02 [-0.64;0.59] | -0.51 **(p=0.06)** [-1.04;0.02] |
|---|---|---|---|---|---|
| Descriptive norm vs. | Personal norm | 2 → 3 | 0.09 [-0.44;0.62] | -0.21 [-0.8;0.37] | -0.31 [-0.8;0.19] |
| N | | | | 1731 | |

\*: $p<0.05$. \*\*: $p<0.01$; 95%-CI in square brackets; superscript letters refer to the effects in the graphs of the main analyses; bold p-values are p-values of coefficients that are not significant when applying heteroskedasticity-robust standard errors (HC1)

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
