# Peer review of "Polarization in Flood Risk Management?"

_EGUsphere, 2025_

## Author Comment (AC1)

**Response to RC1 – Submission NHESS-2025-1362**

Dear Reviewer,

Thank you for taking the time to review our manuscript "Polarization in Flood Risk Management? Sensitivity of norm perception and responsibility attribution to frequent flood experience". We appreciate your evaluation and the insightful comments that help to improve the quality of our work. Below, we address each of your comments point by point and propose corresponding adjustments to the submitted manuscript.

**Comment 1:The introduction frames the literature gap as there being limited knowledge on the drivers of responsibility attribution and norm perception, and suggest frequent flood experience as being a possible driver. Yet, from my understanding quite some studies have looked into the drivers of these two concept , as shown in the literature overview as well. The authors could elaborate how their study extend on previous literature to strengthen their research gap.**

**Answer:** In the literature review, we present existing knowledge and theories on the drivers of norm perception and responsibility attribution. From this we support your impression that studies have already looked at the drivers of responsibility attribution and (to a much lesser extent) norm perception generally. However, we also conclude from the existing literature that there is a lack of knowledge on how experience, specifically *frequent* experience, interacts with these two concepts, emphasized in the literature part:

*Lines 129-130:* "However, the effect of personal experience, exemplarily through social learning and ingroup dynamics, remains widely unclear."

*Lines 133-135:* "(…) studies investigating personal experience as a driver, especially of norm perception, are scarce. In addition, the relationships between experience, perceived norms, and responsibility attribution are rarely integrated in existing theoretical frameworks."

We are open to write this in a less finite tone, by replacing "widely unclear" with "relatively unclear".

Further, we contribute distinct analyses of attributions to different players in the domain of flood risk management: to self, the community, the city and the state, and compare them. Most of the studies that we reviewed were not distinguished in that detail.

From your comment, we understand that we did not highlight these knowledge gaps and the corresponding contributions to existing knowledge sufficiently in the manuscript. Therefore, in the revised version we will emphasize this more.

**Comment 2:Throughout the manuscript locus of control and self-efficacy seems to be used interchangeably, whereas only self-efficacy is included in the independent variables. While both are construct related to control, self-efficacy in behavioral literature refers to one's confidence in being able to perform/implement a measure. Locus of control, on the other hand, refers to one's belief whether they can control the outcome of their life. The authors could examine which concepts they are referring to, and adjust the manuscript accordingly.**

**Answer**: We support the importance of being strict in using technical terminology. In the manuscript, we investigate the effect of perceived self-efficacy and helplessness on the relationships between frequent flood experience, norm perception, and responsibility attribution. Throughout the text, we sometimes refer to these terms by using the term "locus of control". We did not intend to use self-efficacy and locus of control interchangeably, but rather as a term to subsume perceived self-efficacy and expected helplessness during future floods under one umbrella word. We agree that using "locus of control" can be misleading, as the term is its own psychological concept. Therefore, we propose adjusting the three text parts that include the term by not referring to "locus of control" but "perceived control" or "sense of control" instead. We propose to change the following textparts:

*Lines 167 – 168:* "From the theoretical considerations, we hypothesized that locus of control and group identification could play a mediating role."

*Proposed revision:* "From the theoretical considerations, we hypothesized that perceived control of future flood risk and group identification could play a mediating role."

*Lines 475 – 477:* "In conclusion, we observe patterns of polarization with respect to the attribution of responsibility and the perception of norms when people experience multiple flood events, particularly pronounced in certain groups. Based on this, we hypothesize that locus of control and the desire to maintain a capable self-image are drivers of the observed polarization."

*Proposed revision:* "In conclusion, we observe patterns of polarization with respect to the attribution of responsibility and the perception of norms when people experience multiple flood events, particularly pronounced in certain groups. Based on this, we hypothesize that factors related to sense of control over future flood risk and the desire to maintain a capable self-image are drivers of the observed polarization"

*Line 528 – 531:* "For people with low self-efficacy beliefs and low expected controllability of future floods, a polarization between an increase in perceived social norms versus a decrease in moral obligation to participate in collective actions can be observed. From these results, we conclude that locus of control and group identification could have a polarizing effect."

*Proposed revision:* "For people with low self-efficacy beliefs and low expected controllability of future floods, a polarization between an increase in perceived social norms versus a decrease in moral obligation to participate in collective actions can be observed. From these results, we conclude that sense of control over future flood risk and ingroup identification could have a polarizing effect."

**Comment 3**: **Regarding the questions used to elicit the variables, the author could elaborate on the development of these items/scales. Are these validated items/scales based on previous literature, or did the author develop the items themselves?**

**Answer**: We agree that sharing information on the scaling and definition of the items benefits the scientific value of the manuscript. Therefore, we will add this information to the revised manuscript. Specifically, we propose adding information on the scaling (7-point-likert scale) and conceptualization of the items to section 4.2.

**Comment 4: Another concern relates to the representativeness of the sample, especially in relation to the generalizability of the results. Providing the socio-demographics and reflecting on the representativeness would strengthen the study.**

**Answer:** We agree that the idea that presenting the sociodemographic characteristics of the sample increases the generalizability of the results. Therefore, we will include respective information as well as the information at the national level in the revised version of the manuscript.

**Comment 5: Most concerningly, the (polarization in) flood risk management and implications for disaster risk reduction section is written as if it is broadly applicable. Yet social norms seem to be strongly influenced by their local environment, limiting the more general recommendations.**

**Answer**: In the limitations section of the manuscript, we mentioned that the results and interpretations must be seen in light of this, and that replication studies are needed to support the results and drawn conclusions/ stated implications for disaster risk reduction. However, from your comment, we understand that we should also make this clearer in the implications section and formulate it less generalized. We will take this into account in the revision of the current manuscript by adding the assumed regional dependency of norm perception and responsibility attribution also to the implications section.

**Comment 6: In the limitation section, the exclusion of covariates, such as the socio-demographics, and its implications is reflected upon. I am not suggesting that this should be included explicitly, but the reasoning for such methodological decisions could be reflected upon.**

**Answer**: We acknowledge that including more independent variables in the model would most likely increase the explained variance of the outcome variables, thereby improving a general understanding of the drivers behind norm perception and responsibility attribution. Against this backdrop, we agree that in the manuscript we should better reason why we decided against including further independent variables. The decision is mainly based on two reasons. First, there is (to our knowledge) no theoretical framework nor empirical evidence that sociodemographic variables are necessarily impacting responsibility attribution and norm perception that would reason the respective model specification. Second, with the current study, we aim to shed light on a potential relationship between frequent flood experience, norm perception, and responsibility attribution and not to build a model that best predicts the outcome variables. We will add this reasoning to the revised version of the manuscript to section 4.2.

Independently of that, we want to emphasize that including gender, ownership structure, and age as covariates does not alter the effect strength and significance of the experience variable. We are open to adding this information as footnote to the revised manuscript.

**Comment 7: No regression statistics are reported, such as the observations and adjusted r-squared per model iteration.**

**Answer**: We agree that presenting the sample size per model iteration would benefit the study. Therefore, we will add this information to the revised version of the manuscript. However, we do not think that adding the adjusted R-squares and F-statistics to the presented regression statistics would substantially improve the interpretability of the results since the scientific goal of our study is to estimate the link between frequent flood experience, norm perception, and responsibility attribution rather than explaining a high share of variance in the outcome variables. However, we have no strong objection to including the respective values in the revised manuscript in the interest of completeness and transparency.

**Comment 8**: **Line 37, Köhler & Han (2024) should be Köhler and Han (2024)**

**Answer**: We will adjust this in the revised version of the manuscript.

**Comment 9: Line 73, Begg (2017) should be Begg et al. (2017)**

**Answer:** We will adjust this in the revised version of the manuscript.

**Comment 10: Line 106, highlighly, should this be highly?**

**Answer**: This should be "highly" and will be corrected in the revised manuscript.

**Comment 10: Appendix A1 seems to be missing confidence intervals for personal responsibility**

**Answer**: The missing confidence intervals are not intended and will be added in the revised version of the manuscript.

Thank you again for your valuable input and critical assessment of our manuscript. We are looking forward to your feedback on the proposed revisions.

Sincerely,

Lisa on behalf of the co-authors

---

## Author Comment (AC2)

**Response to RC2 – Submission NHESS-2025-1362**

Dear Reviewer,

Thank you for taking the time to review our manuscript *"Polarization in Flood Risk Management? Sensitivity of norm perception and responsibility attribution to frequent flood experience"*. We appreciate your evaluation and the insightful comments that help to improve the quality of our work. Below, we address each of your comments point by point and propose corresponding adjustments to the submitted manuscript.

**Comment 1: The literature review on "perception of norms" covers topics on self-efficacy and ingroup identification, which are the chosen moderating variables in RQ2. This aspect is well-argued. However, the literature review on "attribution of responsibility" only discusses the potential influence of self-efficacy and the influence of ingroup identification is missing.**

**Answer:** In the literature review, we present existing knowledge on the drivers of responsibility attribution and norm perception in general. This part is not intended to provide reasoning for the moderation analyses. Instead, the moderation analyses follow an exploratory approach aimed at understanding the relationships observed in RQ1. From your comment, we understand that we should clarify this better in the manuscript.

In your comment, you specifically refer to ingroup identification as moderator of the relationship between frequent flood experience and responsibility attribution. Here, we assume that the observed variations might be rooted in motivated cognition. We specifically argue that people who highly identify with their community might be particularly likely to ascribe responsibility for flood protection to the public to maintain a positive image of their group despite of potentially insufficient protection levels.

**Comment 2: The literature review and theoretical link can better clarify how self-efficacy and helplessness is connected to perceived control. Or explain how previous literature often associates perceived control with self-efficacy and helplessness, motivating these two measures as moderating variables in the analysis.**

**Answer:** In the manuscript, we subsume self-efficacy beliefs and expected helplessness under the umbrella of sense of control. The reason why we believe that this subsumption is reasonable is the following definitions of the three concepts:

Helplessness: "a state of incapacity, vulnerability, or powerlessness associated with the perception that one cannot do much to improve a negative situation that has arisen" (APA, 2025a).

Self-efficacy: "an individual's subjective perception of their capability to perform in a given setting or to attain desired results" (APA, 2025b).

Perceived control: "subjective belief that one can achieve desired sates and avoid undesired states through one's own efforts" (Dorsch, 2025; translated from German)

The three definitions have in common that they emphasize the (perceived) capacity to shape a certain situation or associated outcomes through own agency. Helplessness specifically refers to a state of powerlessness when facing an external demand, i.e. threatening situation. Self-efficacy highlights the perceived ability to undertake measures to avoid such a situation. From your comment, we understand that we did not make clear enough in the manuscript why we sometimes subsume both under perceived control. We propose emphasizing this more in the revised version and to add the above cited definitions.

**Comment 3: The concept of helplessness is also underexplored in the literature review and theoretical link sections, compared to other key variables of the empirical analysis.**

**Answer:** With the moderation analyses we follow an exploratory approach to better understand the patterns observed in RQ1. The section 2.2 on existing knowledge is mainly focused on drivers of norm perception and responsibility attribution. The theoretical section 2.3 is aimed at delivering reasoning for why we assume that a link between frequent flood experience, norm perception and responsibility attribution might exist.

This is clarified in *lines 135-136* in the current version of the manuscript: "In the following section, we attempt to theoretically reason the hypothesized effect of frequent flood experience on the development of perceived norms and responsibility attribution by combining existing theories and concepts."

From your comment, we understand that we should clarify better that the moderation analyses follow an exploratory approach.

**Comment 4: RQ3 is largely unclear and it is important that there is further explanation. What does it mean when you explore the interaction of your dependent variables (norm and responsibility)? Interaction effects are normally for the independent variables. Furthermore, the methods (regression equation 8) and the results (figure 10 and figure 11) do not seem to answer the research question in a straightforward manner. Can the trends given from RQ1 also be used to answer RQ3?**

**Answer:** The question that we want to answer with RQ3 is whether the observed differences in the development of norm perceptions and responsibility attributions over flood experience (observed in RQ1) vary statistically significantly.

In the current version of the manuscript, we delivered the reasoning for RQ3 in *lines 172-173*: "With the third research question, we aim to uncover possible discrepancies between how much individuals feel responsible and the perceived expectations of their community members regarding private flood protection."

We explained the respective methodological approach in *lines 250-251*: "We adopted the mixed-effects approach to answer RQ3. Instead of distinctly comparing norm/responsibility constructs, we incorporated them into one model (formula 8). We iteratively compared single norm constructs to all responsibility variables."

Specifically, instead of comparing norm constructs/ responsibility attribution separately (RQ1), we created a variable that allowed us to also compare the development of norm

perception and responsibility attribution in one model. From your comment, we understand that we did not make clear enough what specifically we want to answer with RQ3 and how.

In the revised version of the manuscript, we propose the following adjustment of *lines 172-173*: "With the third research question, we aim to uncover possible discrepancies between the respondents' perceived social norms for flood protection and respective responsibility attributions." Further, we try to make the methodological approach clearer and clarify which variables specifically we interact with each other.

**Comment 5**: **It is good practice to include the complete questionnaire as supplementary information if possible. In Section 4.2, it can be clarified how the survey items were formulated: whether questions or statements are standard/validated survey items, have been used in previous survey research, or have been developed by the author themselves.**

**In Table 1, there are three constructs for norms: injunctive, descriptive, and personal. It is the first time these terms appear, which turns out to be important concepts in the discussion section. It may be helpful to introduce these norm constructs in advance, such as whether these distinctions have been recommended in previous literature.**

**Answer:** We agree that this information would benefit the manuscript and its reproducibility. Therefore, in the revised manuscript we will add the full questionnaire to the Appendix. Moreover, in the revised manuscript we will integrate the information that the scaling of the variables is based on the 7-point-likert scale to section 4.2, where we will also add information on how the items were conceptualized. We further understand that we should define the applied norm constructs earlier in the manuscript and propose adding respective definitions to section 2.2 in the part on perception of norms.

**Comment 6**: **It is also good practice to present the descriptive statistics of the sample, besides the main variables that are included in the analysis, for the reader to get a better picture of the sample such as in terms of their socio-demographics. The sample sociodemographic characteristics can also be compared with population characteristics to determine the representativeness of the sample. It should be made clear if this is not possible, mentioning limited data availability or that these information are not elicited in the survey.**

**Answer:** We agree that presenting the sociodemographic characteristics of the sample enables a better understanding of the sample and the applicability of the results to a broader context. Therefore, we will add such information combined with the respective characteristics of the German population to the revised manuscript.

**Comment 7**: **Furthermore, if the survey data provides socio-demographics, the regression models can be improved by including these as control variables. If it is not feasible, it could be mentioned in the mixed effects model with random intercepts whether some part of the respondent characteristics is taken into account by the individual specific intercept, u_j.**

**Answer:** The survey data provide sociodemographic data on the individual level. However, we decided not to integrate these as control variables in the regressions as there is no theoretical framework nor empirical evidence (to our knowledge) supporting the assumption that these would influence norm perception and responsibility attribution. However, to sufficiently answer your question we tested whether the integration of age, gender, and ownership structure would affect the relationship between frequent flood experience and the outcome variables. Here, we observe that homeowners are statistically significantly less likely to attribute responsibility to the city and more likely to attribute responsibility to themselves. However, as these model adjustments do not change the effect of frequent flood experience, neither the effect direction nor its significance, and as our goal is not to build a model explaining as much of the variance in the outcome variables as possible, but to model the link between frequent flood experience, norm perception, and responsibility attribution, we do not support the respective model adjustment in the manuscript. However, we are open to integrate the respective regression tables in the appendix and mention the observed effects as a footnote.

**Comment 8: It may be useful to provide a correlation matrix of the variables listed in Table 1.**

**Answer:** In the revised manuscript, we can integrate this table into the appendix.

**Comment 9**: **Equation 2: The linear equation for norms lacks motivation. Does it serve as a baseline to test if any relationship exists? If so, there should also be a linear equation for responsibility attribution. Only having the linear equation for norms made it seem like it is intentionally for the purpose of showing an increasing relationship for norms (given the statistically significant result, even though the nonlinear effects with dummy variables are statistically insignificant). Do note that a linear equation with FFE as the independent variable may be less appropriate given that a value of 3 indicates experiencing 3 or more floods. FFE is not a continuous variable.**

**Answer:** The motivation for the linear equation for norm perceptions and nonlinear equation for responsibility attributions lies in the descriptive analyses. Here, we observe that the link between frequent flood experience and norm perception follows a rather linear path and the link between frequent flood experience and responsibility attribution a nonlinear path (i.e., changing directions and strengths of the relationship).

In *lines 205-206,* we stated: "For the norm constructs, the descriptive analyses suggest rather linear relationships, which we comparatively tested with a simple linear regression (formula 2)."

From your comment, we understand that we should make clearer in the revised manuscript why we decided in favor of these model specifications and should add the respective reasoning also for responsibility attribution.

**Comment 10**: **The terms "$n\_{ij}$" and "$r\_{ij}$" used in the equations may need to be "$n\_i$" and $r\_i$" instead because these norm and responsibility constructs are categorical/dummy variables that applies to all individuals and do not vary for each individual j.**

**Answer:** It is correct that the indices n_ij and r_ij do not differ between respondents. Instead, they are coded 1-3, respectively, 1-4 for each respondent. However, with this, we follow the proposed notation for mixed models, whereas j represents the group (in our case the respondent) and i the variable nested in that group (in our case the norm/ responsibility constructs). Therefore, we propose keeping that notation.

**Comment 11**: **Potentially crucial to consider: it may be inappropriate to include interaction effects in regression without including the main effects. This applies to equation 3-7. For instance, equation 3 should include n_i (given the correction above) so that each individual j has also different intercepts for the different norm constructs, such that the coefficients of n_i captures the average z_norm scores when FFE=0. The main effect of each FFE may also need to be included. This will make the interaction effect capture the additional differences in marginal effects on top of the main effect, instead of capturing both the main effect and interaction effect when the regression equations are as is currently written. I am also open to further discussing this point.**

**Answer:** We agree that the main effects should be integrated in the estimation to avoid the biases you mentioned. We already considered the main effects in the estimations but did not include them in the regression equations in the previous version. In the revised version of the manuscript, we will adjust the regression equations accordingly.

**Comment 12**: **The regression equation for RQ3 remains unclear. What do each of the nr_ij represent? Is it written with the appropriate notation? It seems that equation 8 estimates equation 3 and 4 jointly, instead of separately. It is unclear how this equation answers the research question RQ3. Looking at the corresponding results in Figure 10 and 11, what is the reason for choosing comparing injunctive norm with own responsibility and descriptive norms with collective responsibility, respectively? Testing these comparisons is not clear given regression equation 8 and RQ3.**

**Answer:** We also tested the other norm constructs and responsibility attributions against each other but only included those in the manuscript as they showed significant differences and target the same "stakeholder-level":

- the respondent's perceived individual responsibility vs. their perception of what others expect them to do
- the respondent's perception of others protective behaviors vs. the respondents perceived collective responsibility.

From your comment, we understand that we should add that reasoning to the revised manuscript.

**Comment 13**: **It could be raised as a limitation that the study treats all previous flood experience equally, not taking account the variations in severity or impact of previous flood experiences. The paper can further reflect upon this limitation, while it is only briefly mentioned in future research section, line 512-513. For example, it could be the case that the nonlinear effects found after the first flood experience are mainly driven by a severe flood, such as the 2002 flood mentioned in line 181-183.**

**Answer:** Whereas we agree that severity of the flood event might explain nonlinear relationships on the individual-level, we assume that such biases do not play a fundamental role in the estimations on the whole sample level. The reason for this assumption is that there are variations among the respondents regarding which flood event was their first, second, or third. For example, whereas for some respondents, the 2002 flood event was the first they were affected by, for others it might have been already the second event. However, in the revised manuscript, we can include this aspect in the discussion section along with our reasoning for why we do not consider it a fundamental limitation.

**Comment 14**: **An interesting direction for future research that can be mentioned in the paper is to survey the theoretical mechanisms that the paper has identified or hinted, but were not explicitly tested. For example, do frequent flood experiences increase social interactions with different actors in FRM (as in line 139-141)? Or does frequent flood experiences change the evaluation of the effectiveness of private protective behaviour (as in line 143-144)?**

**Answer:** In the section "directions for future research", we mention that a path for future research might be to investigate how frequent flood experience relates to "characteristics of the social setting, such as collective efficacy and social cohesion" (*lines 510-511*). Here, we are open to adding social interaction as a further example. Additionally, we support that the investigation of links between frequent flood experience and the assessment of protective behaviors would enrich the stated pool of further research ideas.

**Comment 15**: **In the introduction: Perhaps in line 50 before the sentence, "In Section 2…", also mention that the paper will explore moderating variables (self-efficacy beliefs and ingroup identification), since it is also an insightful finding of the paper.**

**Answer:** We agree that mentioning the moderation analyses in that section would benefit the manuscript.

Therefore, we propose the following adjustment in *line 50:* "To get a better understanding of the observed links between frequent flood experience, norm perception, and responsibility attribution, we performed moderation analyses, taking expected helplessness during future floods, perceived self-efficacy to protect from future flooding and stated identification with other residents as moderators."

**Comment 16: Line 203: What was meant with iteratively adjusted the reference group? It should not matter which one is the reference group. The estimated coefficients for the flood experience dummies should all be relative to the reference group. So, there should not be a need to specify that the regression is estimated separately with different reference group if this is what was meant by iteratively adjusting the reference group.**

**Answer:** We adjusted the reference groups to account for the effect of each additional flood experience separately. For example, to estimate the effect of the second flood event on norm perception, the defined reference group contained people who experienced one flood event. To estimate the effect of the third flood event, the respective reference group contained people who experienced two flood events.

In the manuscript this was described in *lines 203-205*: "We applied single-level linear regression models and iteratively adjusted the reference group to investigate the effect of each flood experience separately. In formula 1, we exemplarily present the equation for investigating the effect of the first flood experience, taking zero flood experience as the reference category."

**Comment 17: Line 245-246: It is also common to test the coefficients against a significance level of 0.01, 0.05, and also 0.1. So, it does not have to be elaborated that p-values that slightly exceed 0.05 will still be interpreted as statistically significant. The coefficients that are significant at a 0.1 level can be interpreted as having "weak evidence", "some evidence", or "marginally significant" for the direction of the relationship.**

**Answer:** The sample size of the applied dataset amounts to 1730 respondents, leading to sample sizes of 1638 to 1721 for estimations without moderators (depending on the outcome variable). Therefore, we decided that p=0.05 is the threshold for statistical significance. However, in the moderation analyses, the sample size is smaller and is even below 100 for some groups. Because of this and since the respective p-values only exceed 0.05 when accounting for heteroskedasticity-consistent standard errors (and only marginally), we decided to relax the chosen threshold a bit and still take the respective relationships into account when interpreting the results.

**Comment 18: Showing the results visually with a line graph is practical. Perhaps, what can be considered is adding the standard errors to the points in the line graph so the readers can more easily infer the statistical significance.**

**Answer:** We agree that including standard errors in the graphs would further assist readers in assessing statistical significance. However, given that the current graphs already present a substantial amount of information and are, in parts, relatively complex to interpret, we are concerned that adding further elements could additionally compromise clarity. Consequently, we would prefer to keep the current presentation.

**Comment 19: Additional regression statistics can be shown in Table 3 and Table 4, such as sample size, f-statistic, and adjusted r-squared. This also applies to the tables in the appendix.**

**Answer:** We support the idea that presenting the sample size for each estimation would benefit the study and agree to include this in the revised manuscript. Generally, we do not believe that presenting adjusted R-squares and F-statistics substantially increases the interpretability of the results or their empirical value as the overarching goal of our study is to model the link between frequent flood experience, norm perception, and responsibility attribution and not to define a model that explains a high share of variance in the outcome variables. However, we do not have a strong opinion against presenting the respective values for the sake of integrity in the revised version of the manuscript.

**Comment 21: Abstract line 16-17: Specify how the gap in responsibility attribution widens. That responsibility attributed to the city/state increases, while responsibility attributed to self/the community increases.**

**Answer:** We will adjust the respective text part in the abstract in *line16*: "(…) as individuals experience multiple flood events, the gap between assigned responsibility to the self/ the community (decrease) vs. the city/the state widens (increase)."

**Comment 22: Abstract line 17-18: Also specify in what way norms are less dynamic. That results show that norms perception weakly increases with frequent flood experience.**

**Answer:** We will adjust the respective text part in the abstract in *line 18*: "(…) are less dynamic. Specifically, variations in effect strength and direction can only be observed for the perception on injunctive norms."

**Comment 23: Line 220: Do not write "n_ij/r_ij" in the text to prevent confusion. It could be initially thought as a fraction. Instead what can be written is "To compare the effects of flood experience between different norm and responsibility constructs, we added n_ij and r_ij as interaction effects, respectively."**

**Answer:** We understand that this notation may cause confusion, and we agree to change it accordingly in the revised version of the manuscript.

**Comment 24: Line 229: Can replace "low self efficacy" with just "self efficacy" for consistency, since the moderating variables are categories of perceived helplessness and self-efficacy.**

**Answer:** We will change the formulation in the revised manuscript to

" (…) (i.e., perceived helplessness, self-efficacy beliefs) (…)"

**Comment 25: Equation 6: the notation of the dependent variable can be consistent with how the dependent variable is written in other notations.**

**Answer:** This notation was unintentional and resulted from an automatic adjustment by the writing software. In the revised version of the manuscript, we will ensure that such automatic adjustment does not occur.

**Comment 26: Tables in the appendix: Instead of writing the p-values that are just above 0.05 in the cells, you can give a note below the tables indicating three significance levels, "*: p<0.1; **: p<0.05; ***: p<0.01".**

**Answer:** As discussed above, we would generally maintain the threshold of $p=0.05$ for statistical significance.

Thank you again for your valuable input and critical assessment of our manuscript. We are looking forward to your feedback on the proposed revisions.

Sincerely,

Lisa on behalf of the co-authors

**References**:

APA (2025a). APA Dictionary of Psychology: Helplessness, last accessed: 02.09.2025. https://dictionary.apa.org/helplessness

APA (2025b): APA Dictionary of Psychology: Self-efficacy, last accessed: 02.09.2025. https://dictionary.apa.org/self-efficacy

Dorsch (2025): Dorsch Lexikon der Psychologie: Kontrollwahrnehmung, last accessed: 02.09.2025. https://dorsch.hogrefe.com/stichwort/kontrollwahrnehmung

---

## Author Response (AR1)

**Answers to the reviewer's comments on the manuscript: "Polarization in Flood Risk Management? Sensitivity of norm perception and responsibility attribution to frequent flood experience"**

**Reviewer I**

| Reviewer Comment | Answer | Implementation |
|---|---|---|
| The introduction frames the literature gap as there being limited knowledge on the drivers of responsibility attribution and norm perception, and suggest frequent flood experience as being a possible driver. Yet, from my understanding quite some studies have looked into the drivers of these two concepts, as shown in the literature overview as well. The authors could elaborate how their study extend on previous literature to strengthen their research gap. | In the literature review, we present existing knowledge and theories on the drivers of norm perception and responsibility attribution. From this we support your impression that studies have already looked at the drivers of responsibility attribution and (to a lesser extent) norm perception generally. However, we also conclude from the existing literature that there is a lack of knowledge on how experience, specifically *frequent* experience, interacts with these two concepts, emphasized in the literature part:

Lines 129-130: "However, the effect of personal experience, exemplarily through social learning and ingroup dynamics, remains widely unclear."
Lines 133-135: "(…) studies investigating personal experience as a driver, especially of norm perception, are scarce. In addition, the relationships between experience, perceived norms, and responsibility attribution are rarely integrated in existing theoretical frameworks."

We are open to write this in a less finite tone, by replacing "widely unclear" with "relatively unclear".

Further, we contribute distinct analyses of attributions to different players in the domain of flood risk management: to self, the community, the city and the state, and compare them. Most of the studies that we reviewed were not distinguished in that detail.

From your comment, we understand that we did not highlight these knowledge gaps and the corresponding contributions to existing knowledge sufficiently in the manuscript. Therefore, in the revised version we will emphasize this more. | L. 141: "However, the effect of personal experience, exemplarily through social learning and ingroup dynamics, remains relatively unclear"

L. 106: "To conclude, there are some studies that have looked at the interplay between hazard experience and responsibility attribution. However, the role played by hazard *frequency* is widely missing. Most of these studies have examined responsibility attribution to either the self or the government. With the current study, we broaden the research lens by also integrating responsibility attributed to the community into our analyses." |
| Throughout the manuscript locus of control and self-efficacy | We support the importance of being strict in using technical terminology. In the manuscript, we investigate the effect of perceived self-efficacy and | L. 178: "Here, we hypothesized that perceived control of future flood risk and group identification could play a |

seems to be used interchangeably, whereas only self-efficacy is included in the independent variables. While both are construct related to control, self-efficacy in behavioral literature refers to one's confidence in being able to perform/implement a measure. Locus of control, on the other hand, refers to one's belief whether they can control the outcome of their life. The authors could examine which concepts they are referring to, and adjust the manuscript accordingly.

helplessness on the relationships between frequent flood experience, norm perception, and responsibility attribution. Throughout the text, we sometimes refer to these terms by using the term "locus of control". We did not intend to use self-efficacy and locus of control interchangeably, but rather as a term to subsume perceived self-efficacy and expected helplessness during future floods under one umbrella word. We agree that using "locus of control" can be misleading, as the term is its own psychological concept. Therefore, we propose adjusting the three text parts that include the term by not referring to "locus of control" but "perceived control" or "sensed control" instead.

Lines 167 – 168:
**Original:** "From the theoretical considerations, we hypothesized that locus of control and group identification could play a mediating role."
**Proposed revision:** "From the theoretical considerations, we hypothesized that perceived control of future flood risk and group identification could play a mediating role."

Line: 475 – 477:
**Original:** "In conclusion, we observe patterns of polarization with respect to the attribution of responsibility and the perception of norms when people experience multiple flood events, particularly pronounced in certain groups. Based on this, we hypothesize that locus of control and the desire to maintain a capable self-image are drivers of the observed polarization."
**Proposed revision:** "In conclusion, we observe patterns of polarization with respect to the attribution of responsibility and the perception of norms when people experience multiple flood events, particularly pronounced in certain groups. Based on this, we hypothesize that factors related to sensed control over future flood risk and the desire to maintain a capable self-image are driver of the observed polarization"

Line 528 – 531:
**Original:** "For people with low self-efficacy beliefs and low expected controllability of future floods, a polarization between an increase in

mediating role. This hypothesis is reasoned in the theoretical considerations presented in section 1.3."

L. 545: "Based on this, we hypothesize that factors related to sensed control over future flood risk and the desire to maintain a capable self-image are drivers of the observed polarization."

L. 605: "For people with low self-efficacy beliefs and low expected controllability of future floods, a polarization between an increase in perceived social norms versus a decrease in moral obligation to participate in collective actions can be observed. From these results, we conclude that sensed control over future flood risk and ingroup identification could have a polarizing effect."

| | perceived social norms versus a decrease in moral obligation to participate in collective actions can be observed. From these results, we conclude that locus of control and group identification could have a polarizing effect."
 **Proposed revision:** "For people with low self-efficacy beliefs and low expected controllability of future floods, a polarization between an increase in perceived social norms versus a decrease in moral obligation to participate in collective actions can be observed. From these results, we conclude that sensed control over future flood risk and ingroup identification could have a polarizing effect." | |
|---|---|---|
| Regarding the questions used to elicit the variables, the author could elaborate on the development of these items/scales. Are these validated items/scales based on previous literature, or did the author develop the items themselves? | We agree that sharing information on the scaling and definition of the items benefits the scientific value of the manuscript. Therefore, we will add this information to the revised manuscript. Specifically, we propose adding information on the scaling (7-point-likert scale) and conceptualization of the items to section 4.2. | L. 215: "Except for the variable indicating the number of experienced flood events, all variables are scaled from 1 to 7, based on the 7-point-likert scale. The Likert-scale is a commonly applied approach to measure perceptions and attitudes in psychological research."

 L. 218: "For a better understanding of the applied psychological constructs and their operationalization, see the definitions presented in Table 3." |
| Another concern relates to the representativeness of the sample, especially in relation to the generalizability of the results. Providing the socio-demographics and reflecting on the representativeness would strengthen the study. | We agree that the idea that presenting the sociodemographic characteristics of the sample increases the generalizability of the results. Therefore, we will include respective information as well as the information at the national level in the revised version of the manuscript. | L. 205: "The survey participants are on average older, more likely to own their place of residence, and more highly educated compared to the general population of the Federal State of Saxony (Table 1). Particularly the differences in the ownership structure might harm the representativeness of the sample regarding the attributions of responsibility. However, this assumption has to be seen against the backdrop of limited existing knowledge on determinants of responsibility attribution for flood risk management, exemplarily the effect of ownership."

 I added a table (Table 1) with some information on the sociodemographic characteristics of the survey sample. |

| | | |
|---|---|---|
| Most concerningly, the (polarization in) flood risk management and implications for disaster risk reduction section is written as if it is broadly applicable. Yet social norms seem to be strongly influenced by their local environment, limiting the more general recommendations. | In the limitations section of the manuscript, we mentioned that the results and interpretations must be seen in light of this, and that replication studies are needed to support the results and drawn conclusions/ stated implications for disaster risk reduction. However, from your comment, we understand that we should also make this clearer in the implications section and formulate it less generalized. We will take this into account in the revision of the current manuscript by adding the assumed regional dependency of norm perception and responsibility attribution also to the implications section. | L. 550: "First, we observe that more flood-experienced people in the sample are less likely to attribute responsibility for flood protection to their community."

L. 554: "Second, the decreasing personal norm to participate in collective actions to reduce flood risk among respondents with low self-efficacy and control beliefs could be targeted by communicating low-barrier collective actions that do not involve a lot of resources."

L. 561: "Importantly, these implications of the current study have to be seen against the backdrop of the perception of social norms and responsibility attributions being most likely influenced by the social environment of the respondents, thereby limiting the generalization of the results." |
| In the limitation section, the exclusion of covariates, such as the socio-demographics, and its implications is reflected upon. I am not suggesting that this should be included explicitly, but the reasoning for such methodological decisions could be reflected upon. | We acknowledge that including more independent variables in the model would most likely increase the explained variance of the outcome variables, thereby improving a general understanding of the drivers behind norm perception and responsibility attribution. Against this backdrop, we agree that in the manuscript we should better reason why we decided against including further independent variables. The decision is mainly based on two reasons. First, there is (to our knowledge) no theoretical framework nor empirical evidence that sociodemographic variables are necessarily impacting responsibility attribution and norm perception that would reason the respective model specification. Second, with the current study, we aim to shed light on a potential relationship between frequent flood experience, norm perception, and responsibility attribution and not to build a model that best predicts the outcome variables. We will add this reasoning to the revised version of the manuscript to section 4.2.

Independently of that, we want to emphasize that including gender, ownership structure, and age as covariates does not alter the effect | L. 278: "The decision to not integrate further explanatory variables, such as sociodemographic characteristics, is predominantly reasoned in the research objective of the current study. The primary aim is to investigate potential relationships between FFE, norm perception, and responsibility attribution instead of specifying a model that explains a huge share of the variance in the outcome variables. Further, there is (to our knowledge), no theoretical framework nor substantial empirical evidence suggesting that sociodemographic variables play a crucial role in norm perception and responsibility attribution, reasoning their integration in the regression model."

Footnote: "Integrating age, gender and ownership structure as covariates to the model does not alter the effect strength and significance of FFE." |

| | strength and significance of the experience variable. We are open to adding this information as footnote to the revised manuscript. | |
|---|---|---|
| No regression statistics are reported, such as the observations and adjusted r-squared per model iteration. | We agree that presenting the sample size per model iteration would benefit the study. Therefore, we will add this information to the revised version of the manuscript. However, we do not think that adding the adjusted R-squares and F-statistics to the presented regression statistics would substantially improve the interpretability of the results since the scientific goal of our study is to estimate the link between frequent flood experience, norm perception, and responsibility attribution rather than explaining a high share of variance in the outcome variables. However, we have no strong objection to including the respective values in the revised manuscript in the interest of completeness and transparency. | Added to the Table 5 and Table 6 |
| Line 37, Köhler & Han (2024) should be Köhler and Han (2024) | We will adjust this in the revised version of the manuscript. | Adjusted |
| Line 73, Begg (2017) should be Begg et al. (2017) | We will adjust this in the revised version of the manuscript. | Adjusted |
| Line 106, highlighly, should this be highly? | This should be "highly" and will be corrected in the revised manuscript. | Adjusted |
| Appendix A1 seems to be missing confidence intervals for personal responsibility | The missing confidence intervals are not intended and will be added in the revised version of the manuscript. | Adjusted |

**Reviewer II**

| Reviewer Comment | Answer | Implementation |
|---|---|---|
| The literature review on "perception of norms" covers topics on self-efficacy and ingroup identification, which are the chosen moderating variables | In the literature review, we present existing knowledge on the drivers of responsibility attribution and norm perception in general. This part is not intended to provide reasoning for the moderation analyses. Instead, the moderation analyses follow an exploratory approach aimed at | L. 177: "To understand the patterns observed in RQ1 and link them theoretically, we exploratively tested for boundary conditions for the effect of FFE on perceived norms and responsibility attributions. Here, we hypothesized that perceived control of future flood risk |

| | | |
|---|---|---|
| in RQ2. This aspect is well-argued. However, the literature review on "attribution of responsibility" only discusses the potential influence of self-efficacy and the influence of ingroup identification is missing. | understanding the relationships observed in RQ1. From your comment, we understand that we should clarify this better in the manuscript.

In your comment, you specifically refer to ingroup identification as moderator of the relationship between frequent flood experience and responsibility attribution. Here, we assume that the observed variations might be rooted in motivated cognition. We specifically argue that people who highly identify with their community might be particularly likely to ascribe responsibility for flood protection to the public to maintain a positive image of their group despite of potentially insufficient protection levels. | and group identification could play a mediating role. This hypothesis is reasoned in the theoretical considerations presented in section 2.3" |
| The literature review and theoretical link can better clarify how self-efficacy and helplessness is connected to perceived control. Or explain how previous literature often associates perceived control with self-efficacy and helplessness, motivating these two measures as moderating variables in the analysis. | In the manuscript, we subsume self-efficacy beliefs and expected helplessness under the umbrella of sensed control. The reason why we believe that this subsumption is reasonable is the following definitions of the tree concepts:

Helplessness: "a state of incapacity, vulnerability, or powerlessness associated with the perception that one cannot do much to improve a negative situation that has arisen" (APA, 2025).

Self-efficacy: "an individual's subjective perception of their capability to perform in a given setting or to attain desired results" (APA, 2025).

Perceived control: "subjective belief that one can achieve desired sates and avoid undesired states through one's own efforts" (Dorsch, 2025; translated from German)

The three definitions have in common that they emphasize the (perceived) capacity to shape a certain situation or associated outcomes through own agency. Helplessness specifically refers to a state during a challenging situation. Self-efficacy highlights the perceived ability to undertake measures to avoid such a situation. From your comment, we understand that we did not make clear enough in the manuscript why we sometimes subsume both under perceived control. We propose emphasizing this more in the revised version and to add the above cited definitions. | L. 223: "By integrating the variables "self-efficacy" and "helplessness" as moderators in the analysis, we aim to shed light on the role of perceived control in shaping norm perception and responsibility attribution when individuals experience multiple flood events. This subsumption under perceived control is reasoned in both factors emphasizing the (perceived) capacity to shape a certain situation or associated outcomes through own agency (see Table 3). Helplessness specifically refers to a state during a threatening situation. Self-efficacy, in turn, highlights the perceived ability to undertake measures to avoid such a situation. To obtain interpretable classes for the moderation analyses, we built three groups for each moderator and assigned respondents based on their responses to the statements (Table 4)." |

| | | |
|---|---|---|
| The concept of helplessness is also underexplored in the literature review and theoretical link sections, compared to other key variables of the empirical analysis. | With the moderation analyses we follow an exploratory approach to better understand the patterns observed in RQ1. The section 2.2 on existing knowledge is mainly focused on drivers of norm perception and responsibility attribution. The theoretical section 2.3 is aimed at delivering reasoning for why we assume that a link between frequent flood experience, norm perception and responsibility attribution might exist.

This is clarified l. 135-136 in the current version of the manuscript:

"In the following section, we attempt to theoretically reason the hypothesized effect of frequent flood experience on the development of perceived norms and responsibility attribution by combining existing theories and concepts."

From your comment, we understand that we should clarify better that the moderation analyses follow an exploratory approach. | L. 177: "To understand the patterns observed in RQ1 and link them theoretically, we exploratively tested for boundary conditions for the effect of FFE on perceived norms and responsibility attributions. Here, we hypothesized that perceived control of future flood risk and group identification could play a mediating role. This hypothesis is reasoned in the theoretical considerations presented in section 2.3." |
| RQ3 is largely unclear and it is important that there is further explanation. What does it mean when you explore the interaction of your dependent variables (norm and responsibility)? Interaction effects are normally for the independent variables. Furthermore, the methods (regression equation 8) and the results (figure 10 and figure 11) do not seem to answer the research question in a straightforward manner. Can the trends given from RQ1 also be used to answer RQ3? | The question that we want to answer with RQ3 is whether the observed differences in the development of norm perceptions and responsibility attributions over flood experience (observed in RQ1) vary statistically significantly. In the current version of the manuscript, we delivered the following reasoning for RQ3 l.172-173:

"With the third research question, we aim to uncover possible discrepancies between how much individuals feel responsible and the perceived expectations of their community members regarding private flood protection"

We explained the respective methodological approach in l. 250-251:

"We adopted the mixed-effects approach to answer RQ3. Instead of distinctly comparing norm/responsibility constructs, we incorporated them into one model (formula 8). We iteratively compared single norm constructs to all responsibility variables." | L. 185: "With the third research question, we aim to uncover possible discrepancies between the respondents' perceived social norms for flood protection and respective responsibility attributions:"

L. 310: "To answer RQ3, we adjusted the mixed-effects model (formula (3) and (4)) . Instead of analyzing norm perceptions and responsibility attributions separately, we combined them into a single model by introducing the interaction variable $nr_{ij}$ and the outcome variable $z\_NORM\_RESPONSIBILITY_{ij}$ (formula 8). The variable $nr_{ij}$ specifies which norm or responsibility construct serves as the reference category against which the remaining constructs are compared. This model setup allows us to contrast how FFE interacts with norm perceptions versus responsibility attributions within the same model, enabling us to draw conclusions on statistically significant differences between them. We |

| | | |
|---|---|---|
| | Specifically, instead of comparing norm constructs/ responsibility attribution internally (RQ1), we created a variable that allowed us to also compare the development of norm perception and responsibility attribution in one model. From your comment, we understand that we did not make clear enough what specifically we want to answer with RQ3 and how. In the revised version of the manuscript, we propose the following adjustment of l.172-173:

"With the third research question, we aim to uncover possible discrepancies between the respondents' perceived social norms for flood protection and respective responsibility attributions"

Further, we try to make the methodological approach clearer and clarify which variables specifically we interact with each other. | iteratively compared single norm constructs to all responsibility variables." |
| It is good practice to include the complete questionnaire as supplementary information if possible. In Section 4.2, it can be clarified how the survey items were formulated: whether questions or statements are standard/validated survey items, have been used in previous survey research, or have been developed by the author themselves.
In Table 1, there are three constructs for norms: injunctive, descriptive, and personal. It is the first time these terms appear, which turns out to be important concepts in the discussion section. It may be helpful to introduce these norm constructs in advance, such as whether | We agree that this information would benefit the manuscript and its reproducibility. Therefore, in the revised manuscript we will add the full questionnaire as Supplementary File. Moreover, in the revised manuscript we will integrate the information that the scaling of the variables is based on the 7-point-likert scale to section 4.2, where we will also add information on how the items were conceptualized. We further understand that we should define the applied norm constructs earlier in the manuscript and propose adding respective definitions to section 2.2 in the part on perception of norms. | L. 215: "Except for the variable indicating the number of experienced flood events, all variables are scaled from 1 to 7, based on the 7-point-likert scale. The Likert-scale is a commonly applied approach to measure perceptions and attitudes in psychological research."

L. 111: "Social norms can be classified into descriptive and injunctive norms. Descriptive norms are defined as the (perceived) actual behavior of others (APA, 2025a) and injunctive norms as the perception of how people should behave (APA, 2025e)." |

| these distinctions have been recommended in previous literature. | | |
|---|---|---|
| It is also good practice to present the descriptive statistics of the sample, besides the main variables that are included in the analysis, for the reader to get a better picture of the sample such as in terms of their socio-demographics. The sample sociodemographic characteristics can also be compared with population characteristics to determine the representativeness of the sample. It should be made clear if this is not possible, mentioning limited data availability or that these information are not elicited in the survey. | We agree that presenting the sociodemographic characteristics of the sample enables a better understanding of the sample and the applicability of the results to a broader context. Therefore, we will add such information combined with the respective characteristics of the German population to the revised manuscript. | L. 205: "The survey participants are on average older, more likely to own their place of residence, and more highly educated compared to the general population of the Federal State of Saxony (Table 1). Particularly the differences in the ownership structure might harm the representativeness of the sample regarding the attributions of responsibility. However, this assumption has to be seen against the backdrop of limited existing knowledge on determinants of responsibility attribution for flood risk management, exemplarily the effect of ownership."

I added Table 1 containing information on the sociodemographic characteristics of the sample. |
| Furthermore, if the survey data provides socio-demographics, the regression models can be improved by including these as control variables. If it is not feasible, it could be mentioned in the mixed effects model with random intercepts whether some part of the respondent characteristics is taken into account by the individual specific intercept, $u\_j$. | The survey data provide sociodemographic data on the individual level. However, we decided not to integrate these as control variables in the regressions as there is no theoretical framework nor empirical evidence (to our knowledge) supporting the assumption that these would influence norm perception and responsibility attribution. However, to sufficiently answer your question we tested whether the integration of age, gender, and ownership structure would affect the relationship between frequent flood experience and the outcome variables. Here, we observe that homeowners are statistically significantly less likely to attribute responsibility to the city and more likely to attribute responsibility to themselves. However, as these model adjustments do not change the effect of frequent flood experience, neither the effect direction nor its | L. 278: "The decision to not integrate further explanatory variables, such as sociodemographic characteristics, is predominantly reasoned in the research objective of the current study. The primary aim is to investigate potential relationships between FFE, norm perception, and responsibility attribution instead of specifying a model that explains a huge share of the variance in the outcome variables. Further, there is (to our knowledge), no theoretical framework nor substantial empirical evidence suggesting that sociodemographic variables play a crucial role in norm perception and responsibility attribution, reasoning their integration in the regression model." |

| | | |
|---|---|---|
| | significance, and as our goal is not to build a model explaining as much of the variance in the outcome variables as possible, but to model the link between frequent flood experience, norm perception, and responsibility attribution, we do not support the respective model adjustment in the manuscript. However, we are open to integrate the respective regression tables in the appendix and mention the observed effects as a footnote. | Footnote: "Integrating age, gender and ownership structure as covariates to the model does not alter the effect strength and significance of FFE. We observe that homeowners are statistically significantly less likely to attribute responsibility to the city and more likely to attribute responsibility to themselves." |
| It may be useful to provide a correlation matrix of the variables listed in Table 1. | In the revised manuscript, we can integrate this table into the appendix. | Integrated as Appendix A |
| Equation 2: The linear equation for norms lacks motivation. Does it serve as a baseline to test if any relationship exists? If so, there should also be a linear equation for responsibility attribution. Only having the linear equation for norms made it seem like it is intentionally for the purpose of showing an increasing relationship for norms (given the statistically significant result, even though the nonlinear effects with dummy variables are statistically insignificant). Do note that a linear equation with FFE as the independent variable may be less appropriate given that a value of 3 indicates experiencing 3 or more floods. FFE is not a continuous variable. | The motivation for the linear equation for norm perceptions and nonlinear equation for responsibility attributions lies in the descriptive analyses. Here, we observe that the link between frequent flood experience and norm perception follows a rather linear path and the link between frequent flood experience and responsibility attribution a nonlinear path (i.e., changing directions and strengths of the relationship).

In l. 205-206, we stated:
"For the norm constructs, the descriptive analyses suggest rather linear relationships, which we comparatively tested with a simple linear regression (formula 2)."

From your comment, we understand that we should make clearer in the revised manuscript why we decided in favor of these model specifications and should add the respective reasoning also for responsibility attribution. | L. 240: "For responsibility attribution, the descriptive analyses suggest a nonlinear relationship, which we accounted for by estimating the effect of each flood experience individually. Exemplarily, formula 1 presents the equation for investigating the effect of the first flood experience taking zero flood experience as the reference category. For the norm constructs, the descriptive analyses suggest rather linear relationships, reasoning the linear model specification presented in formula 2." |

| | | |
|---|---|---|
| The terms "n_ij" and "r_ij" used in the equations may need to be "n_i" and r_i" instead because these norm and responsibility constructs are categorical/dummy variables that applies to all individuals and do not vary for each individual j. | It is correct that the indices n_ij and r_ij do not differ between respondents. Instead, they are coded 1-3, respectively, 1-4 for each respondent. However, with this, we follow the proposed notation for mixed models, whereas j represents the group (in our case the respondent) and i the variable nested in that group (in our case the norm/ responsibility constructs). Therefore, we propose keeping that notation. | Kept like it is |
| Potentially crucial to consider: it may be inappropriate to include interaction effects in regression without including the main effects. This applies to equation 3-7. For instance, equation 3 should include n_i (given the correction above) so that each individual j has also different intercepts for the different norm constructs, such that the coefficients of n_i captures the average z_norm scores when FFE=0. The main effect of each FFE may also need to be included. This will make the interaction effect capture the additional differences in marginal effects on top of the main effect, instead of capturing both the main effect and interaction effect when the regression equations are as is currently written. I am also open to further discussing this point. | We agree that the main effects should be integrated in the estimation to avoid the biases you mentioned. We considered the main effects in the estimations but did not include them in the regression equations. In the revised version of the manuscript, we will adjust the regression equations accordingly. | Adjusted |

| | | |
|---|---|---|
| The regression equation for RQ3 remains unclear. What do each of the nr_ij represent? Is it written with the appropriate notation? It seems that equation 8 estimates equation 3 and 4 jointly, instead of separately. It is unclear how this equation answers the research question RQ3. Looking at the corresponding results in Figure 10 and 11, what is the reason for choosing comparing injunctive norm with own responsibility and descriptive norms with collective responsibility, respectively? Testing these comparisons is not clear given regression equation 8 and RQ3. | We also tested the other norm constructs and responsibility attributions against each other but only included those in the manuscript as they showed significant differences and target the same "stakeholder-level":

- the respondent's perceived individual responsibility vs. their perception of what others expect them to do
- the respondent's perception of others protective behaviors vs. the respondents perceived collective responsibility.

From your comment, we understand that we should add that reasoning to the revised manuscript. | L. 310: "To answer RQ3, we adjusted the mixed-effects model (formula (3) and (4)) . Instead of analyzing norm perceptions and responsibility attributions separately, we combined them into a single model by introducing the interaction variable $nr_{ij}$ and the outcome variable $z\_NORM\_RESPONSIBILITY_{ij}$ (formula 8). The variable $nr_{ij}$ specifies which norm or responsibility construct serves as the reference category against which the remaining constructs are compared. This model setup allows us to contrast how FFE interacts with norm perceptions versus responsibility attributions within the same model, enabling us to draw conclusions on statistically significant differences between them."

L. 444: "The remaining combinations of comparable norm perception and responsibility attribution did not show statistically significant diverging developments." |
| It could be raised as a limitation that the study treats all previous flood experience equally, not taking account the variations in severity or impact of previous flood experiences. The paper can further reflect upon this limitation, while it is only briefly mentioned in future research section, line 512-513. For example, it could be the case that the nonlinear effects found after the first flood experience are mainly driven by a severe flood, such as the 2002 flood mentioned in line 181-183. | Whereas we agree that severity of the flood event might explain nonlinear relationships on the individual-level, we assume that such biases do not play a fundamental role in the estimations on the whole sample level. The reason for this assumption is that there are variations among the respondents regarding which flood event was their first, second, or third. For example, whereas for some respondents, the 2002 flood event was the first they were affected by, for others it might have been already the second event. However, in the revised manuscript, we can include this aspect in the discussion section along with our reasoning for why we do not consider it a fundamental limitation. | L. 578: "Lastly, the nonlinear relationships could be linked to specific characteristics of distinct flood events (e.g., variations in severity). However, whereas we cannot rule out that such bias exists on the individual level, we do not assume that it does play a crucial role in the estimations on the whole sample. The reason for this assumption is that there are variations among the respondents regarding which flood event was their first, second, or third. For example, whereas for some respondents, the 2002 flood event was the first they were affected by, for others it might have been already the second event." |

| | | |
|---|---|---|
| An interesting direction for future research that can be mentioned in the paper is to survey the theoretical mechanisms that the paper has identified or hinted, but were not explicitly tested. For example, do frequent flood experiences increase social interactions with different actors in FRM (as in line 139-141)? Or does frequent flood experiences change the evaluation of the effectiveness of private protective behaviour (as in line 143-144)? | In the section "directions for future research", we mention that a path for future research might be to investigate how frequent flood experience relates to "characteristics of the social setting, such as collective efficacy and social cohesion" (l. 510-511). Here, we are open to adding social interaction as a further example. Additionally, we support that the investigation of links between frequent flood experience and the assessment of protective behaviors would enrich the stated pool of further research ideas. | L. 588: "Linking other characteristics of the social setting, such as collective efficacy, social cohesion and interaction to FFE could provide further insight into how changing hazard contexts affect communities." |
| In the introduction: Perhaps in line 50 before the sentence, "In Section 2…", also mention that the paper will explore moderating variables (self-efficacy beliefs and ingroup identification), since it is also an insightful finding of the paper. | We agree that mentioning the moderation analyses in that section would benefit the manuscript. Therefore, we propose the following adjustment in line 50:

"To get a better understanding of the observed links between frequent flood experience, norm perception, and responsibility attribution, we performed moderation analyses, taking expected helplessness during future floods, perceived self-efficacy to protect from future flooding and stated identification with other residents as moderators." | L. 53: "To get a better understanding of the observed links between frequent flood experience, norm perception, and responsibility attribution, we performed moderation analyses, taking expected helplessness during future floods, perceived self-efficacy to protect from future flooding and identification with other residents as moderators." |
| Line 203: What was meant with iteratively adjusted the reference group? It should not matter which one is the reference group. The estimated coefficients for the flood experience dummies should all | We adjusted the reference groups to account for the effect of each additional flood experience separately. For example, to estimate the effect of the second flood event on norm perception, the defined reference group contained people who experienced one flood event. To estimate the effect of the third flood event, the respective reference group contained people who experienced two flood events. In the manuscript this was described in l. 203-205: | Kept like it is |

| | | |
|---|---|---|
| be relative to the reference group. So, there should not be a need to specify that the regression is estimated separately with different reference group if this is what was meant by iteratively adjusting the reference group. | "We applied single-level linear regression models and iteratively adjusted the reference group to investigate the effect of each flood experience separately. In formula 1, we exemplarily present the equation for investigating the effect of the first flood experience, taking zero flood experience as the reference category." | |
| Line 245-246: It is also common to test the coefficients against a significance level of 0.01, 0.05, and also 0.1. So, it does not have to be elaborated that p-values that slightly exceed 0.05 will still be interpreted as statistically significant. The coefficients that are significant at a 0.1 level can be interpreted as having "weak evidence", "some evidence", or "marginally significant" for the direction of the relationship. | The sample size of the applied dataset amounts to 1730 respondents, leading to sample sizes of 1638 to 1721 for estimations without moderators (depending on the outcome variable). Therefore, we decided that p=0.05 is the threshold for statistical significance. However, in the moderation analyses, the sample size is smaller and is even below 100 for some groups. Because of this and since the respective p-values only exceed 0.05 when accounting for heteroskedasticity-consistent standard errors (and only marginally), we decided to relax the chosen threshold a bit and still take the respective relationships into account when interpreting the results. | Kept like it is |
| Showing the results visually with a line graph is practical. Perhaps, what can be considered is adding the standard errors to the points in the line graph so the readers can more easily infer the statistical significance. | We agree that including standard errors in the graphs would further assist readers in assessing statistical significance. However, given that the current graphs already present a substantial amount of information and are, in parts, relatively complex to interpret, we are concerned that adding further elements could additionally compromise clarity. Consequently, we would prefer to keep the current presentation. | Kept like it is |
| Additional regression statistics can be shown in Table 3 and Table 4, such as sample size, f-statistic, and adjusted r-squared. This also applies to the tables in the appendix. | We support the idea that presenting the sample size for each estimation would benefit the study and agree to include this in the revised manuscript. Generally, we do not believe that presenting adjusted R-squares and F-statistics substantially increases the interpretability of the results or their empirical value as the overarching goal of our study is to model the link between frequent flood experience, norm perception, and responsibility attribution and not to define a model that explains a high | Adjusted |

| | share of variance in the outcome variables. However, we do not have a strong opinion against presenting the respective values for the sake of integrity in the revised version of the manuscript. | |
|---|---|---|
| Abstract line 16-17: Specify how the gap in responsibility attribution widens. That responsibility attributed to the city/state increases, while responsibility attributed to self/the community increases. | We will adjust the respective text part in the abstract in l. 16:

"(…) as individuals experience multiple flood events, the gap between assigned responsibility to the self/ the community (decrease) vs. the city/the state widens (increase)." | L. 18: "Second, we detect a diverging trend among respondents who experienced multiple flood events, with greater responsibility attributed to public authorities and less to their own communities." |
| Abstract line 17-18: Also specify in what way norms are less dynamic. That results show that norms perception weakly increases with frequent flood experience. | We will adjust the respective text part in the abstract in l. 18:

"(…) are less dynamic. Specifically, variations in effect strength and direction can only be observed for the perception on injunctive norms." | L. 16: "Changes in norm perceptions are less dynamic. Specifically, variations in effect strength and direction can only be observed for the perception of injunctive norms." |
| Line 220: Do not write "n_ij/r_ij" in the text to prevent confusion. It could be initially thought as a fraction. Instead what can be written is "To compare the effects of flood experience between different norm and responsibility constructs, we added n_ij and r_ij as interaction effects, respectively." | We understand that this notation may cause confusion, and we agree to change it accordingly in the revised version of the manuscript. | L. 263: "To compare the effects of flood experience between different norm/responsibility constructs, we added $n_{ij}$ $and$ $r_{ij}$ as interaction effect (formula 3,4)." |
| Line 229: Can replace "low self efficacy" with just "self efficacy" for consistency, since the moderating variables are | We will change the formulation in the revised manuscript to

" (…) (i.e., perceived helplessness, self-efficacy beliefs) (…)" | L. 284: "RQ2: Do factors related to perceived control of future flood risk (i.e., perceived helplessness, self-efficacy beliefs) and social connectedness (i.e., group identification) moderate the relationship between frequent |

| | | |
|---|---|---|
| categories of perceived helplessness and self-efficacy. | | flood experience, norm perception, and responsibility attribution?" |
| Equation 6: the notation of the dependent variable can be consistent with how the dependent variable is written in other notations. | This notation was unintentional and resulted from an automatic adjustment by the writing software. In the revised version of the manuscript, we will ensure that such automatic adjustment does not occur. | Adjusted |
| Tables in the appendix: Instead of writing the p-values that are just above 0.05 in the cells, you can give a note below the tables indicating three significance levels, "*: p<0.1; **: p<0.05; ***: p<0.01". | As discussed above, we would generally maintain the threshold of p=0.05 for statistical significance. | Kept like it is |

---

## Author Response (AR2)

**Dear Dr. Sven Fuchs,**

thank you for handling our manuscript "Polarization in Flood Risk Management? Sensitivity of norm perception and responsibility attribution to frequent flood experience", your valuable feedback and criticism. In the following, we will address each of your comments and state how we adjusted the manuscript accordingly.

**Comment:** Include regression results for RQ1 that incorporate key socio-demographic controls in the appendix. These results should help demonstrate that your preferred models remain stable and that the magnitude and significance of the coefficients are largely unaffected.

**Answer:** In the revised version of the manuscript, we added regression results under consideration of sociodemographic covariates to the appendix (see Table B1 and B2). Namely, age, gender, tenure, and educational background of the respondents.

As only eight people indicated to identify as diverse, we solely focused on the difference between males and females in the regression analysis, which we also added as a note to the regression tables.

For the education variables, we decided to adopt binary variables to obtain the effect of each level of education. It could also have been a way to treat education as a linear variable. However, we assume that the gaps between the different educational levels are not equal. For example, it is not clear what kind of education respondents obtained who indicated not to have any degree. The fact that a proportion of the respondents participated in a different schooling system as they were educated in the German Democratic Republic (GDR) makes it even harder to assess whether the differences between the groups vary or not. Because of this, we decided that treating education categorically is statistically the most appropriate way to consider educational background.

A huge share of respondents did either not answer at all how much income their household has (N=156) or did not want to share that piece of information (N=431), implying a substantial reduction of the whole sample size when considering income in the regression. This also leads to an unequal reduction of each experience group by more than 100 respondents. As integrating the income variable to the regression models suggests that it does not have a significant effect on responsibility attributions and norm perceptions, we belief that this systematic exclusion of respondents is not reasoned in further knowledge acquisition. Although we believe that there is also value in understanding why so many respondents did not answer or did not want to share their household income in an anonymous survey, this is beyond the scope of the current study. Because of this, we prefer to not include the income variable in the current study's appendix.

In summary, integrating gender, age, and tenure does not alter the effect of flood experience. The same applies to the educational background of the respondents. The adjusted R-square increases marginally (in absolute terms). Consequently, the models remain largely stable.

**Comment:** Include socio-demographic characteristics in Table A1 (correlation matrix). Key variables such as gender, age, homeownership, education/qualification, and income should be added. For multi-category variables, please use simplified binary indicators (e.g., homeowner vs. non-homeowner; university degree vs. no university degree).

**Answer:** We added age, gender, tenure, educational background, and income to the correlation tables (see Table A2).

**Comment:** If you retain the preferred models that exclude socio-demographic characteristics, please adjust the structure of the explanation provided. The rationale for excluding these controls (lines 266–271) should be introduced earlier in the section – specifically, immediately after the sentence ending in "…of FFE comparable." at line 230 – because the justification applies to all models in the manuscript, not solely to those addressing RQ1.

**Answer:** We replaced the rationale to lines 230-236.

In addition to these key adjustments proposed by the editor, we also considered the following comments by the two reviewers:

Reviewer 1:

**Comment:** Regarding the presented education levels in table 1, it would be recommended to match the German levels to their English translation/equivalent to enhance clarity for non-German readers.

**Answer:** We added English translations (see Table 1 in combination with footnote 1) . However, as the germen schooling systems has some particularities and schools that do not exist internationally, these translations only serve as a rough orientation.

Reviewer 2:

**Comment:** It was mentioned that the descriptive analyses of the effect for norm perceptions is linear, making the linear model preferred. Why is it that in Table 6, the effects of each flood event as dummies is still reported? Was a similar regression to equation (1) also performed? Then this should also be reported in section 4.3 analytical steps.

**Answer:** We kept reporting the dummy variables to support the statement that the relationship between frequent flood experience and norm perception follows a linear path (i.e., the dummies are, except for one, not statistically significant, but the linear coefficients are). We added the following sentence to 4.3: "For comparison reasons, we also present the outcomes when applying the regression model in (1) for norm perception. " (lines 243-244).

**Comment:** Furthermore, for the analysis of norm perceptions, the linear model and the dummy variable model of FFE are two different regressions, so should they have different F-statistics? The regression statistics was reported in the same columns in Table 6, making it seem as if both the linear and dummy variables were estimated simultaneously. I actually prefer for equation (2) to be modelled the same way as equation (1) for consistency with other models (equation 3-8 are all with dummy variables), and that Figure 4 is shown as if FFE is estimated as discrete dummy variables.

**Answer:** We understand that the previous table was misleading. Therefore, we adjusted the table and present separate columns for the regression with dummy variables and the linear model. We prefer formula (2) to be a linear model as this is suggested by the descriptive analysis and also by the outcomes of the regression (i.e., significance of the coefficients within the linear model).

**Comment:** It would be helpful to include a translated English version of the survey, given that NHESS has an international audience.

**Answer:** The key variables of the study, i.e., norm perception and flood experience, have been translated in the methods section. Consequently, the questions and measurements that are crucial for understanding and replicating the current study are understandable by an international audience.

We hope that we have now addressed all major comments sufficiently and look forward to receiving your feedback on the revised version of the manuscript.

Kind regards,

Lisa Köhler, on behalf of all co-authors